**Methane in Zackenberg Valley, NE Greenland: Multidecadal growing season fluxes of a high Arctic tundra**

Johan H. Scheller[1,2], Mikhail Mastepanov[1,3], Hanne H. Christiansen[2], Torben R. Christensen[1,3]

[1]Department of Ecoscience, Arctic Research Centre Aarhus University, Roskilde, Denmark

[2]Arctic Geology Department, The University Centre in Svalbard, Longyearbyen, Norway

[3]Oulanka research station, University of Oulu, Finland

*Correspondence to*: Johan H. Scheller (jscheller@ecos.au.dk)

**Abstract.** The carbon balance of high-latitude terrestrial ecosystems plays an essential role in the atmospheric
concentration of trace gases, including carbon dioxide ($CO_2$) and methane ($CH_4$). Increasing atmospheric methane
levels have contributed to ~20 % of the observed global warming since the pre-industrial era. Rising temperatures
in the Arctic are expected to promote the release of methane from Arctic ecosystems. Still, existing methane flux
measurement efforts are sparse and highly scattered, and further attempts to assess the landscape fluxes over
multiple years are needed.

Here we combine multi-year July–August methane flux monitoring (2006–2019) from automated flux chambers in
the central fens of Zackenberg Valley, northeast Greenland, with several flux measurement campaigns on the most
common vegetation types in the valley to estimate the landscape fluxes over 14 years. Methane fluxes based on
manual chamber measurements are available from campaigns in 1997, 1999–2000, and in shorter periods from
2007–2013 and were summarized in several published studies. The landscape fluxes are calculated for the entire
valley floor and a smaller subsection of the valley floor, containing the productive fen area, Rylekærene.

When integrated for the valley floor, the estimated July–August landscape fluxes were low compared to the single
previous estimate, while the landscape fluxes for Rylekærene were comparable to previous estimates. The valley
floor was a net methane source during July–August, with estimated mean methane fluxes ranging from 0.18 to 0.67
mg m$^{-2}$ h$^{-1}$. The mean methane fluxes in the fen-rich Rylekærene were substantially higher, with fluxes ranging
from 0.98 to 3.26 mg m$^{-2}$ h$^{-1}$.

A 2017–2018 erosion event indicates that some fen and grassland areas in the center of the valley are becoming
unstable following pronounced fluvial erosion and a prolonged period of permafrost warming. Although such
physical disturbance in the landscape can disrupt the current ecosystem–atmosphere flux patterns, even pronounced
future erosion of ice-rich areas is unlikely to impact methane fluxes on a landscape scale significantly. Instead,
projected changes in future climate in the valley play a more critical role. The results show that multi-year landscape
methane fluxes are highly variable on a landscape scale and stress the need for long-term spatially distributed
measurements in the Arctic.

## 1 Introduction

It was recognized early in climate change research that the Arctic is particularly prone to increasing temperatures,
and considerable changes to ecosystems at high latitudes were to be expected (IPCC, 1990). For decades, hence,

tundra ecosystems have been subject to attention concerning changing climate. Further, high latitude wetlands have, for a long time, been recognized as a contributor to the global atmospheric budget of methane ($CH_4$). In the first global budget, the tundra source was an estimated 1.3 to 13 Tg $y^{-1}$ of a total atmospheric burden of 529 to 825 Tg $y^{-1}$ (Ehhalt, 1974). This term and the overall atmospheric budget have changed remarkably little during nearly 50 years of research, with emission estimates for all global wetlands ranging from 140 to 280 Tg $y^{-1}$ in the 1970s as well as in the 2010s (Christensen, 2014). This still also forms the background for the concern that with Arctic ecosystems warming, the so far modest wet tundra emissions may increase and start a positive feedback in the climate system (Knoblauch et al., 2018). In addition to Arctic warming, landscape changes such as permafrost thaw affecting tundra ecosystems and lake formations have been identified as possible hotspots for increased emissions (Schuur et al., 2015; Walter Anthony et al., 2018). Also, coastal and further offshore marine sources are subject to possible changed emissions in the Arctic (Shakhova et al., 2014; Thornton et al., 2020), but these are not dealt with here. Overall, these concerns led to early and still continuing efforts at quantifying more closely the Arctic natural emissions and their sensitivity and dynamics in relation to climate change.

Studies of tundra methane emissions in several parts of the Arctic were initiated between the 1970s and 1990s (Svensson and Rosswall, 1984; Whalen and Reeburgh, 1988; Morrissey and Livingston, 1992; Christensen, 1993). Zackenberg Valley in northeast Greenland was one of the first high Arctic sites to be added to the circumpolar map of methane flux studies in 1997 (Christensen et al., 2000) after the start of the Zackenberg Ecological Research Operations (ZERO) in 1995 (Meltofte et al., 2008). Since then, the valley has seen several different methane flux studies, and methane monitoring became part of the Greenland Ecosystem Monitoring (GEM) program in 2007 (Mastepanov et al., 2013). With numerous short-term research projects, the monitoring has led to the availability of a unique, large number of years with observations of fluxes compared to any other location in the Arctic.

The overarching background for the sizeable emissions in the Arctic is that waterlogged undisturbed soil environments host stable anaerobic environments with optimal conditions for methanogenic activity. Compared with tropical wetlands influenced heavily by the seasonality of flooding (Nisbet et al., 2019), the arctic source areas tend to be more stable geographically. Their emissions subject to the balance between the production at depth and the microbial oxidation in the aerobic surface layer. However, through recent decades, many factors such as nutrients, plant species composition, topography/hydrology have been found to influence the size of the emissions (AMAP, 2015). From a landscape perspective, the constantly emitting wet soil environments are surrounded by and intermixed with uplands, glaciers, lakes, and rivers, all with their distinct and, in some cases, very different methane flux characteristics. It is rare that a comprehensive analysis of both these small-scale controls and spatial heterogeneity is possible simply due to a lack of locations where enough studies of different kinds of land cover have been conducted. Zackenberg Valley is here unique, with such a wide range of studies available.

Here we compile all methane flux studies conducted so far in Zackenberg valley with the objectives to 1) review the combined information from these studies on temporal and spatial variability of methane fluxes in a composite

high Arctic landscape and 2) assess the sensitivity of the measured fluxes as they respond to climate warming or local changes. A large number of studies and multiple years of observations, along with the addition of flux measurements from a recent gully, provide a unique opportunity to disentangle the effect of different processes and quantify their relative influence on the fluxes on a landscape scale. These processes can broadly be grouped as 1) climate variability and the projections for a gradual warming, 2) increased erosion of vegetated surfaces.

The challenge is to quantify the sensitivity of the landscape fluxes to these factors individually to allow for quantitative analysis of how the factors combine and compare in terms of sensitivity to established climate warming scenarios. To conduct this study, we use a combination of published data and new measurements of methane fluxes from a recently eroded gully near the Zackenberg Research Station and its immediate surroundings.

## 2 Materials and methods

### 2.1 Site description

#### 2.1.1 Study area

Zackenberg Valley is located at 74.47° N, 20.55° W in northeast Greenland. The study area follows the boundaries used in Søgaard et al. (2000) and covers the core monitoring and research projects area of the GEM program (Zone 1A) below 200 m above sea level, encompassing a total area of ~16 km$^2$ on the valley floor (Fig. 1). The vegetated areas consist of continuous and hummocky fens, grasslands, *Salix* snowbeds, and *Cassiope* and *Dryas* heaths. Common species in the fens include *Dupontia psilosantha* and *Eriophorum scheuchzeri*, and graminoids, including *Arctagrostis latifolia*, *Eriophorum triste,* and *Alepecurus alpinus* covers the grassland areas. (Bay, 1998). Rylekærene partly covers the valley floor study area and consists of a 1.3 km$^2$ patterned wet tundra ecosystem dominated by fen areas and divided by drier patches of heaths and grasslands in the central part of the valley (Tagesson et al., 2013). Rylekærene has been a subject of several methane flux studies since the mid-1990s, and several sites within the area have been in use (Fig. 1).

A backward erosion gully, previously referred to as 'thermokarst' (Christensen et al., 2020b), extends from Gadekæret, a small fen area close to Zackenberg Research Station in the central part of the valley (Fig. 1). The dendritically shaped gully formed rapidly during July 2018, most likely triggered by substantial lateral erosion of up to 4.7 m in an outer bend of the river after a glacier lake outburst flood (GLOF) in August 2017 (Tomczyk et al., 2020). The gully is up to 1 to 4 m deep, up to 8 m wide, and ~50 m long in the longest flow direction. This process is different from thermokarst and is similar to the active development of a gully in the northern part of Zackenberg Valley in 1999 (Christiansen et al., 2008). The retrogressive, branching pattern of the gully likely occurs along with ice wedges in the area. Meltwater from a large snowpack in Gadekæret in early 2018 has most likely saturated the thawed sediments and thermally eroded the ice wedges, causing sediment transport from the site and into the river.

The gully was monitored closely during the 2019 field season, but its development had ceased. Standing water covered the bottom of the gully during August 2018, and elevated methane concentrations were detected in the area (Christensen et al., 2020b). In 2019, the bottom of the gully had dried out, and methane concentrations were no longer elevated.

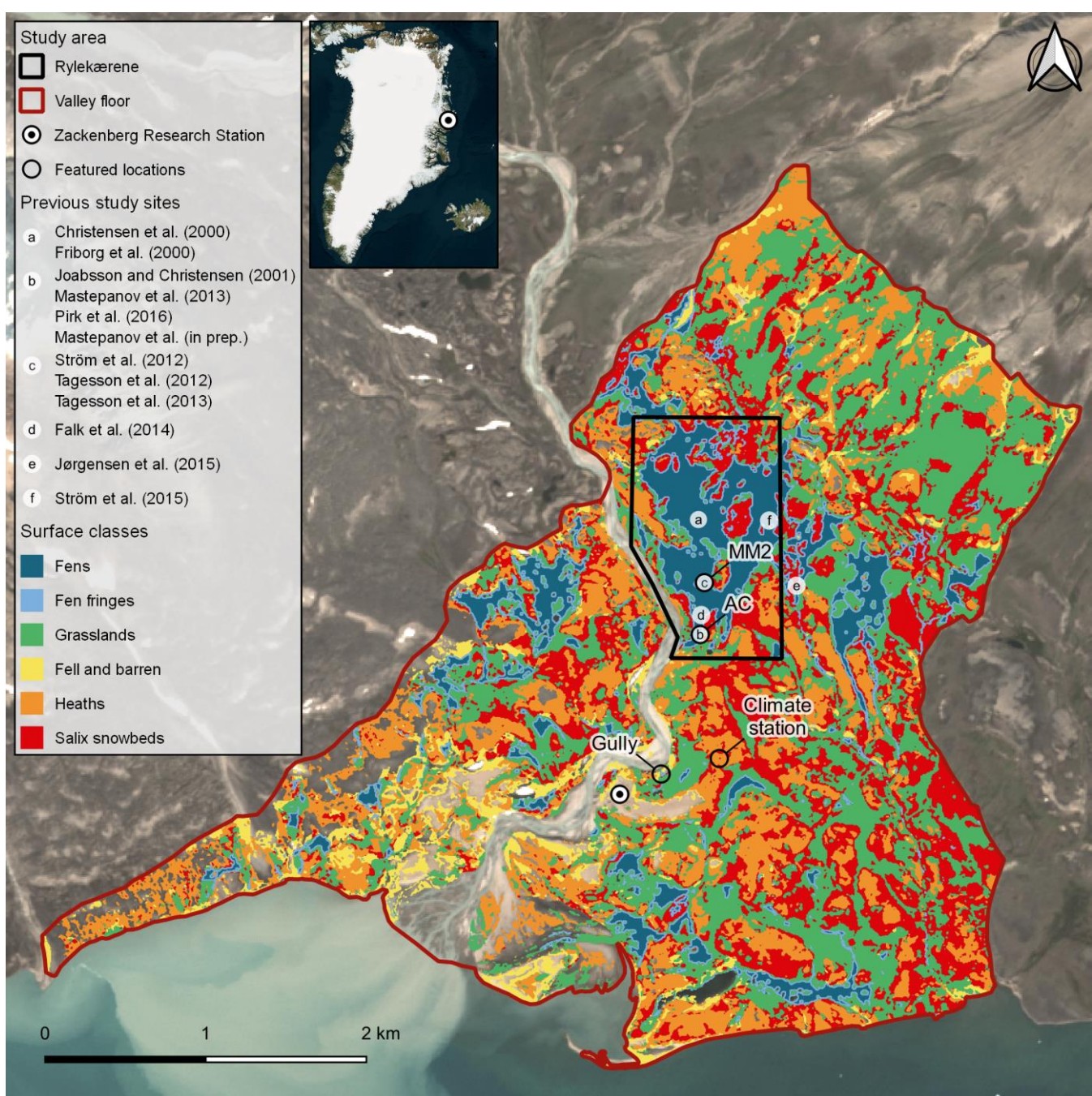

**Figure 1. The study area with dark red and black boundaries showing the valley floor study area and Rylekærene in Zackenberg (74.47° N, 20.55° W) and approximate locations of previous study sites are marked with letters. The two gas flux monitoring sites, AC and MM2, are found at the same places as study sites b and c, marked with black circles. Colored areas show the spatial distribution of the six surface classes used in this study.** *Data sources: HyMap hyperspectral imaging campaign (7 August 2000) (Elberling et al., 2008), ESA Copernicus Sentinel-2 (16 July 2019), and © BING Maps 2021.*

Table 1 shows the coverage of each surface class compared to the valley floor and Rylekærene areas in both absolute and relative terms, based on hyperspectral aerial data from (Elberling et al., 2008), which improved on a

manual land cover classification by Bay (1998). Zackenberg River has changed its course since 2000 after multiple GLOF events (Søndergaard et al., 2015; Tomczyk and Ewertowski, 2020), and we adjusted the land cover classification map to fit the extent of the river in a 2014 orthomosaic of the entire study area (COWI, 2015).

**Table 1. Absolute and relative area distributions for the six main surface cover classes in the two study areas shown in Fig. 1, updated to the Zackenberg River extent in 2014. The 'fen fringe' surface class covers the outer 10 m of the fen areas.**

| Surface class | Valley floor in $m^2$ (% of the area) | Rylekærene in $m^2$ (% of the area) |
|---|---|---|
| **Fens** | 983,920  (6.2 %) | 464,816 (36.6 %) |
| **Fen fringes** | 920,504  (5.8 %) | 176,502 (13.9 %) |
| **Grassland** | 4,429,921 (27.9 %) | 176,135 (13.9 %) |
| **Fell and barren** | 964,643  (6.1 %) | 7,854  (0.1 %) |
| **Heaths** | 3,434,361 (21.6 %) | 138,692 (10.9 %) |
| *Salix* **snowbeds** | 3,244,606 (20.4 %) | 265,979 (20.9 %) |
| *Other* | 2,891,691 (18.1 %) | 41,325  (3.7 %) |
| **Total** | 15,905,003  (100 %) | 1,271,303 (100 %) |

### 2.1.2 Climatology

The study area is defined as high Arctic (Meltofte and Rasch, 2008), with mean annual air temperature and precipitation of −8.6 °C and 253 mm (2008–2018) (López-Blanco et al., 2020). Continuous permafrost underlies the valley (Christiansen et al., 2008), and the active layer (AL) reaches a maximum depth of 0.58 to 0.85 m, increasing at a rate of 0.74 cm year$^{-1}$ (1996–2019) at the ZEROCALM-1 site (Christensen et al., 2020a) near the climate station at the valley floor.

## 2.2 Methane flux measurements

### 2.2.1 Measurements in Rylekærene

The first methane flux measurements in Rylekærene were carried out in 1997 in Christensen et al. (2000) and Friborg et al. (2000). They used manual flux chamber measurements and the eddy covariance (EC) method in the center of the fen area (Fig. 1, site a). In the south end of Rylekærene, Joabsson and Christensen (2001) measured methane fluxes in 1999–2000 (Fig. 1, site b). An automated chamber setup was added nearby in 2005. The automated chambers (AC) measure methane and $CO_2$ fluxes along a topographic gradient from a wet fen to a heath area. The AC site is a part of the GeoBasis subprogram of GEM (Mastepanov et al., 2008; Mastepanov et al., 2013). Tagesson et al. (2012) measured methane and $CO_2$ fluxes by gradient EC methods at a site in the center of Rylekærene in 2008 and 2009. This site (Fig. 1, site c) is located ~250 m north of the AC site and was later promoted to a permanent EC $CO_2$ flux installation for measurements in the GeoBasis subprogram under the MM2 (MicroMet 2) name (Lund et al., 2008–2011; Stiegler et al., 2016).

In 2007, Tagesson et al. (2013) measured methane fluxes of the most common vegetation types in a ~600 m$^2$ area in the center of Rylekærene, 50 to 150 m east of the current location of MM2.

More recently, several studies focused on areas outside the permanent AC and MM2 sites, including Falk et al. (2014) (Fig. 1, site d), and Jørgensen et al. (2015) (Fig. 1, site e) toward the eastern border of Rylekærene. Ström

et al. (2015) measured fluxes in between the AC site and MM2 (Fig. 1, site f). The length of these campaigns, their onset compared to the beginning of the growing season, the sampling area, and strategy vary between studies. Figure 2 shows a timeline of methane flux campaigns from published studies and data from the GEM database.

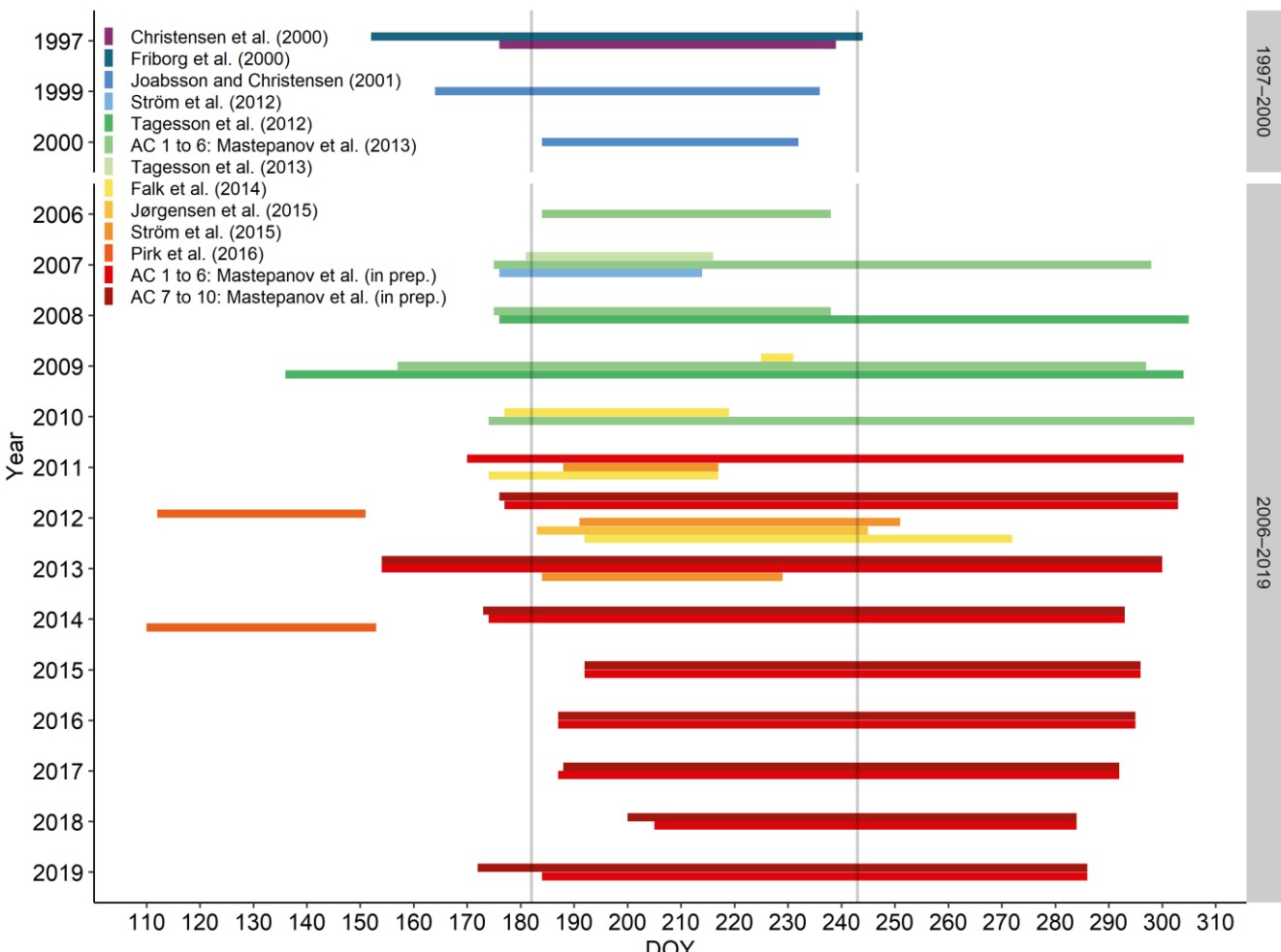

**Figure 2. Timeline of methane measurement campaigns and monitoring in individual years, shown as the day of year (DOY). Colors show different publications, and the two vertical grey bars show 1 July and 31 August in non-leap years. Please note that the years 2001–2005 are left out, as no published studies focusing on methane fluxes in undisturbed areas were made in this period.**

Most of the previous studies in the research area were based on mobile flux chambers, and stationary automatic chambers, which utilized changes in methane concentration measured over time inside a closed chamber to estimate surface fluxes. The chamber was sealed off to all sides but the bottom during each measurement, isolating the flux estimates to a specific time and area, analogous to the approach described by, e.g., Crill et al. (1988) and Livingston and Hutchinson (1995). Methods vary between the studies, with differences in gas analyzers, chamber designs, measurement time, replication numbers, sampling frequency, and the length of the study periods. Details on the methods are summarized in Table 2.

**Table 2. Flux chamber measurements in Zackenberg Valley vary in several ways between studies. Please see each study for a detailed description of the methods. Only the data used in this study are included in the table, i.e., unmanipulated and control plots.**

| Publication and study year | Gas analysis | Chamber design | Sampling time | Plots | Measurement frequency |
|---|---|---|---|---|---|

| | | | | | |
|---|---|---|---|---|---|
| **Christensen et al. (2000) in 1997** | Duplicate syringe sample in two 10 to 15 min intervals. Analysis with a Shimadzu GC14-B gas chromatograph | 13 to 30 L, aluminum dark chambers on bases permanently inserted to 10 to 20 cm depth | 20 to 30 min | 30 plots in five sites | Two times per week |
| **Joabsson and Christensen (2001) in 1999–2000** | Duplicate syringe samples were taken at regular intervals. Analysis with a Shimadzu GC14-B gas chromatograph | 14 to 22 L, aluminum dark chambers on permanently installed bases (15 cm depth) | NA | Six plots in one site (only control plots) | One to three times per week |
| **Ström et al. (2012) in 2007** | LGR DLT200 analyzer | 2.2 L, transparent chamber on permanently installed aluminum bases (set-up[1]) | 3 min | 15 | Two to four days interval |
| **Mastepanov et al. (2013) in 2006–2010, Mastepanov et al. (in prep.) in 2011** | LGR DLT100 model 908-0007 | ~108 L permanently installed transparent, automatic chambers | 5 min | Four (2006) Six (2007–2011) | Every hour |
| **Tagesson et al. (2013) in 2007** | LGR DLT200 analyzer | 10 L, transparent chamber inserted into ~2 cm notches in the ground | 3 min | 55 plots in seven sites | Two to three times per week |
| **Falk et al. (2014) in 2009–2012** | LGR DLT200 analyzer and Gasmet Dx 40-30 Fourier transform infrared spectrometer | 41 L, transparent and dark chambers on bases permanently installed to 15 cm depth | 3 to 7 min | 15 plots in one site (only unmanipulated plot) | Approximately two times per week |
| **Jørgensen et al. (2015) in 2012** | LGR DLT100 analyzer | Volume NA, Transparent and dark chambers on permanent bases inserted to ~10 cm depth | 10 min | 40 plots in four sites | NA, 280 samples in total during four campaigns |
| **Ström et al. (2015) in 2011–2013** | Gasmet Dx 40-30 Fourier transform infrared spectrometer | 41 L, transparent and dark chambers on bases permanently installed to 15 cm depth | 3 to 7 min | 16 to 20 | Approximately two times per week |
| **Pirk et al. (2016b) in 2012 and 2014, Mastepanov et al. (in prep.) in 2012 to 2019** | LGR Greenhouse Gas Analyzer model 908-0011 | ~108 L permanently installed transparent automatic chambers | 5 min | 10 | Every 90 minutes |

For the aims of the current study, we combined existing data of growing season methane from the studies shown in Fig. 2 from 2006 and forward. Only parts of the AC flux time series were used in this study; the July–August flux includes the peak of the growing season in the valley, and it matches the timing of most of the previous studies.

### 2.2.2 Measurements in the gully area

We conducted 113 measurements from 23 August 2019 to 1 September 2019 in a grid covering the recent gully and extending toward a small lake in the Gadekæret fen area, close to Zackenberg Research Station (Fig. 4). The measurements took place between 9 a.m. and 7 p.m. on 43 plots located 4 to 5 m from each other in the three main surface classes in the area: fens, grasslands, and in the gully. Several flux measurements at each surface class were performed every day.

A metal collar was carefully installed on the ground on each plot; a dark acrylic chamber was placed over the collar for a minimum of five minutes. The footprint of the collar was $0.07 \text{ m}^2$; the height from the soil surface to the top of the chamber was recorded for each measurement. The height ranged from 0.26 m to 0.48 m, depending on the surface topography. The chamber was equipped with a fan inside and had a 3 mm vent on the side. A gas analyzer (Ultraportable Greenhouse Gas Analyzer, Los Gatos Research, USA) was connected to the chamber with a pair of 15 m long HDPE tubes. The gas analyzer was continuously measuring methane concentration in the chamber headspace at 1 Hz frequency. After each sample, the chamber was ventilated for at least two minutes until the methane concentration was down to ambient concentration.

## 2.3 Data processing

### 2.3.1 Environmental changes

Monitoring of air (2 m above ground) and soil temperature (0.20 m below ground) were summarized from 1997–2019 data (60 min resolution) from the nearby climate station (Fig. 1), operated by the GEM ClimateBasis program. Mean temperatures are calculated for July–August data (Fig. 3a). The day of 20 % snow cover in the central valley is available from Pedersen et al. (2016) for the 1999–2014 period (Fig. 3b). Day of 20 % snow cover is estimated from GeoBasis monitoring data for 2015–2019. Soil moisture data at the heath site Mix-1, monitored under the GeoBasis program, were summarized as mean soil moisture percentage with SD for July–August (Fig. 3c). Water level data were collected manually daily at the AC site chamber 1 from 2006–2019. Automatic water level data is collected for 2010–2019, with a gap in data in 2013 at two sites, one near chamber 1 and another near chamber 6, positioned slightly higher in the terrain. The mean water levels are calculated for July–August (Fig. 3d), and SD is omitted to improve the readability of the figure. Dataset specifications are available from the links under 'Data and code availability'.

### 2.3.2 Measurements from automated and manual chambers

Data from several sources were included in the calculation of a timeline of landscape fluxes in Zackenberg Valley. Previously published and unpublished data were compiled to estimate fluxes and SE on the six surface types (Table 3), combining mobile flux measurements and flux measurements from AC. The measurement methods are described in detail in their respective publication, but for the AC, additional steps were added after applying the same approach as Mastepanov et al. (2013). The AC flux time series was separated into two datasets: one representing a long time series from a 10 m wide transition zone at the fen fringe (chambers 1 to 6). In this area, chambers with higher numbers are generally drier. Another time series represents the changes since 2012 in the four outer chambers, located further out into the fen (chambers 7 to 10). The flux data do not show a diurnal pattern for July and August, so all available data were used in those two months. The flux measurements were first averaged for each chamber. Mean chamber fluxes were then further averaged into a single mean methane flux for each year for all the six inner chambers. The same was done for the four outer chambers from 2012–2019.

In 2006, only four chambers were operating, and no data were available for chambers 4 and 6. Potential differences in methane flux are corrected by multiplying the mean of the available 2006 data by a coefficient based on 2007 data. This coefficient was found by dividing the mean in chambers 1, 2, 3, and 5 by the mean of all six chambers.

#### 2.3.2.1 Fen flux

The mean July–August flux for chambers 7 to 10 were then combined with the mean growing season fluxes from the other studies listed in Table 3. In the valley-wide vegetation cover map, hummocky and continuous fen were not separated into different classes, even though mean fluxes differ substantially for these two surface types in, e.g., Table 3 in Tagesson et al. (2013) and Table 1 in Christensen et al. (2000).

Each mean flux measured in fen areas was paired with the mean flux measured at the fen fringe. Using R (R Core Team, 2021), we applied unweighted Deming linear regression on the data ($n = 18$, *jackknife method, error ratio*

= 0.44, *Pearson's r* = 0.64, *p*-value threshold = 0.05). The approach accounts for uncertainties in both the fen fringe data and in the fen data to estimate a time series for the fen surface types for 2006–2019. The measurements used in the regression for the fen areas are summarized in Table 3.

### 2.3.2.2 Fen fringe flux

The mean flux from the fen fringes was estimated from the mean July–August flux measured every year in the 2006–2019 period using all the available data from chamber 1 to 6 (Table 3). The timeline of mean fluxes at the fen fringe represents the outer 10 m of all fen surfaces in the valley, shown in Fig. 1 without further processing.

### 2.3.2.3 Grassland flux

The grassland fluxes are held constant over the time series, with data from Tagesson et al. (2013) as input to the 225 calculation (Table 3), while grassland fluxes from Christensen et al. (2000) are omitted due to high spatial variability and a higher average flux.

### 2.3.2.4 Fell and barren fluxes

Jørgensen et al. (2015) found a significant methane uptake on unvegetated fell and barren surfaces in Zackenberg Valley in 2012. These measurements were grouped into a broader 'dry tundra' class that includes *Dryas* heath. 230 Here, we use the mean methane flux from the 'dry tundra' class but only applying it for the fell and barren areas. The mean flux from these surfaces is held constant in our landscape flux time series.

### 2.3.2.5 Heaths and *Salix* fluxes

The heath class in this study includes both *Cassiope*, *Dryas* and *Vaccinium* areas, and data from Christensen et al. (2000), Tagesson et al. (2013), and Jørgensen et al. (2015) are combined to calculate an average estimate for fluxes 235 in these areas, which are held constant over time. Jørgensen et al. (2015) used different groups in their study, where different types of heath areas fall into 'dry' and 'moist' tundra (*Cassiope* and *Salix* heath). We calculated the weighted mean flux for each study for the heath class, and then we calculated a mean and pooled SE. Likewise, the *Salix* snowbed class was calculated in the same way as with heath, with data from the same three studies, but with only the flux from 'moist' tundra from Jørgensen et al. (2015).

**Table 3. Summary of data and processing used in the calculation of landscape methane fluxes for the six surface classes.**

| Surface class | Data source | Measurement year(s) | Processing |
|---|---|---|---|
| **Fens** | Ström et al. (2012) (*n* = 210) | 2007 | Fit mean fluxes to fen fringes time series using Deming linear regression, paired by year, SE estimates from jackknife method |
| | Tagesson et al. (2012) | 2008–2009 | |
| | Tagesson et al. (2013) (*n* = 162) | 2007 | |
| | Falk et al. (2014) (*n* = 35–85) | 2010–2012 | |
| | Ström et al. (2015) (*n* = 80–140) | 2011–2013 | |
| | Mastepanov et al. (in prep.) (*n* = 1,465–3,432) | 2012–2019 | |
| **Fen fringes** | Mastepanov et al. (2013) (*n* = 3,713–8,238) | 2006–2010 | July–August means for chamber 1–6, SE calculated from variability in flux measurements |
| | Mastepanov et al. (in prep.) (*n* = 1,888–4,837) | 2010–2019 | |
| **Grasslands** | Tagesson et al. (2013) (*n* = 110) | 2007 | Constant mean flux of grasslands, SE calculated from reported SD and *n* |
| **Fell and barren** | Jørgensen et al. (2015) (*n* ~ 140) | 2012 | Constant mean flux' dry tundra', SE calculated from reported SD and *n* |
| **Heaths** | Christensen et al. (2000) (*n* ~ 90) | 1997 | Constant mean flux of *Dryas*/*Cassiope*/*Vaccinium* heaths, and 'moist'/'dry' tundra flux, first weighted average by study, pooled SE |
| | Tagesson et al. (2013) (*n* = 162) | 2007 | |
| | Jørgensen et al. (2015) (*n* = 280) | 2012 | |
| ***Salix* snowbeds** | Christensen et al. (2000) (*n* ~ 42) | 1997 | Constant mean flux of all *Salix* snowbed surfaces, and 'moist tundra' flux, pooled SE |
| | Tagesson et al. (2013) (*n* = 51) | 2007 | |

### 2.3.3 Combining data into landscape flux time series

The six combined surface classes occupy most of the study areas (Rylekærene and the valley floor, Table 1). This
static areal coverage of the surface classes was calculated using QGIS v. 3.18.1 (QGIS.org, 2021).

The area-weighted flux was calculated for the two study areas, the valley floor, and Rylekærene, with total areas
of 15,905,003 $m^2$ and 1,271,303 $m^2$. This approach assumes no methane flux in the remaining 18.1 % and 3.7 %
of the study areas, including the river, delta, abrasion plateaus, and boulder fields in the 'other' category in Table
1. The area-weighted landscape flux is calculated for each year, with fluxes in the fens and fen fringes being the
only to change over time.

### 2.3.4 Fluxes in the gully area

In the recently eroded gully area, the fluxes in 2019 were calculated using the ordinary least square linear (OLS)
regression described in Pirk et al. (2016a). Of the 113 measurements, 102 had a significant ($p < 0.05$) regression
slope. The remaining 11 measurements were found on both the grassland areas and on recently eroded surfaces.
They showed an increase in concentration over time close to zero, and they are interpreted as areas with no flux.
The 11 measurements are included in the calculation of the mean flux of the recently eroded surfaces in the gully.
The mean flux and the standard error (SE) for all measurements were calculated by calculating the mean flux for
repeated measurements for each plot. Further averaging for their respective surface class (gully, grasslands, fens)
was done afterward, showing how fluxes can change in an area after an erosion event. Flux measurements from the
gully area are limited to 10 days in the late growing season.

### 2.3.5 Valley flux and future impacts on methane emissions with increasing erosion activity

Recent active riverbank erosion in Zackenberg Valley raises the question of how the methane flux on a valley-scale
will change with increasing erosion in the future, and the sensitivity to such changes is explored below.

#### 2.3.5.1 Changes in methane flux from increasing temperatures

In their study, Geng et al. (2019) compared the potential methane flux in the Zackenberg area modeled for the late
21[st] century with present-day conditions. They found an exponential growing season temperature–methane flux
relationship based on Zackenberg and Kobbefjord data from 2008–2015. Present-day temperatures were modeled
for the 1991–2010 period, and late 21[st] century temperatures were modeled for 2081–2100 under the Representative
Concentration Pathway 8.5 (RCP8.5). Both present-day and future mean temperatures were modeled using the
ECHAM5 general circulation model to limit the cold bias in the model, and the relative increase in methane flux
was +141 % (Geng, personal communication, 24 April 2020). The relative increase ranges from +114 % to +171
% when considering the lower and upper 95 % confidence bounds, see Fig. 3 in Geng et al. (2019).

We assume a linear increase in valley methane flux of +141 % for this sensitivity study, ranging from +114 % to +171 % from 2016–2100, even though this period does not fully match the modeled periods, i.e., from 1991–2010 to 2081–2100.

### 2.3.5.2 Sensitivity to increasing erosion activity

We establish three pathways to quantify how erosion could affect the landscape flux in Zackenberg Valley. In the first pathway, we calculate the impact on the mean valley flux if the eroded areas are growing at an annual rate equivalent to the size of the recent gully (720 $m^2$ $y^{-1}$). In the second and third pathway, the eroded area starts at 720 $m^2$ every year and grows to 5 and 10 times that area per year, respectively, i.e., a linear increase over 85 years from 720 $m^2$ $y^{-1}$ to 3,600 $m^2$ $y^{-1}$, and from 720 $m^2$ $y^{-1}$ to 7,200 $m^2$ $y^{-1}$. In this calculation, the erosion can only occur in zones with excessive ice-rich permafrost near rivers and streams. To identify these zones, we use GIS data available from Cable et al. (2018). Inside our study area (Fig. 1), we identify parts of the landscape that likely have 'excess ice-rich top 1 m permafrost' and are located near 'rivers', which include both rivers and tributaries, see Fig. 12 in Cable et al. (2018). Based on observations from the recent gully, erosion can occur up to 50 m from the rivers. Inside the identified zones, we model erosion of the size specified above and the fractional cover of surface classes. The reduction in mean flux for the eroded areas is subtracted from the landscape flux. As time goes, the eroded areas increase, causing larger impacts on the landscape flux, while all fluxes increase at the same rate as when the mean temperatures increase.

### 2.3.5.3 Revegetation of eroded areas

A similar gully in the northern part of the valley developed in 1999 (Christiansen et al., 2008), and it shows signs of revegetation on ~40 % of the eroded areas to grassland after 20 years. This estimate is based on a visual interpretation of 100 points randomly scattered over eroded parts within the gully in an orthophoto from August 2019. From the estimate, we set the regrowth rate equal to 2 % $year^{-1,}$ and this rate of regrowth is included in the overall calculation.

## 3 Results

### 3.1 Environmental changes

Figure 3a shows the development of July–August soil and air temperatures since 1997 at 0.2 m depth and 2 m height measured at the climate station. The mean July–August temperature is 5.9 °C for air temperatures and 4.7 °C for the available soil temperatures. Air temperatures increased by 0.07 °C $year^{-1}$ (1997–2019, $n = 23$, *Pearson's* $r = 0.43$, $p = 0.04$), while no significant linear trend is observed in the soil temperature data during the same period. The timing of snowmelt in the study area (Fig. 3b) shows substantial variations between years, from 30 (DOY 150) to 27 July (DOY 208). During July and August, there is so far no clear trend toward drier or wetter conditions for both the heath areas (Fig. 3c) and the fens at the AC site (Fig. 3d) from the mid-2000s and forward. Measurements from chamber 6 show generally drier conditions than from chamber 1, which is located further out in the fen.

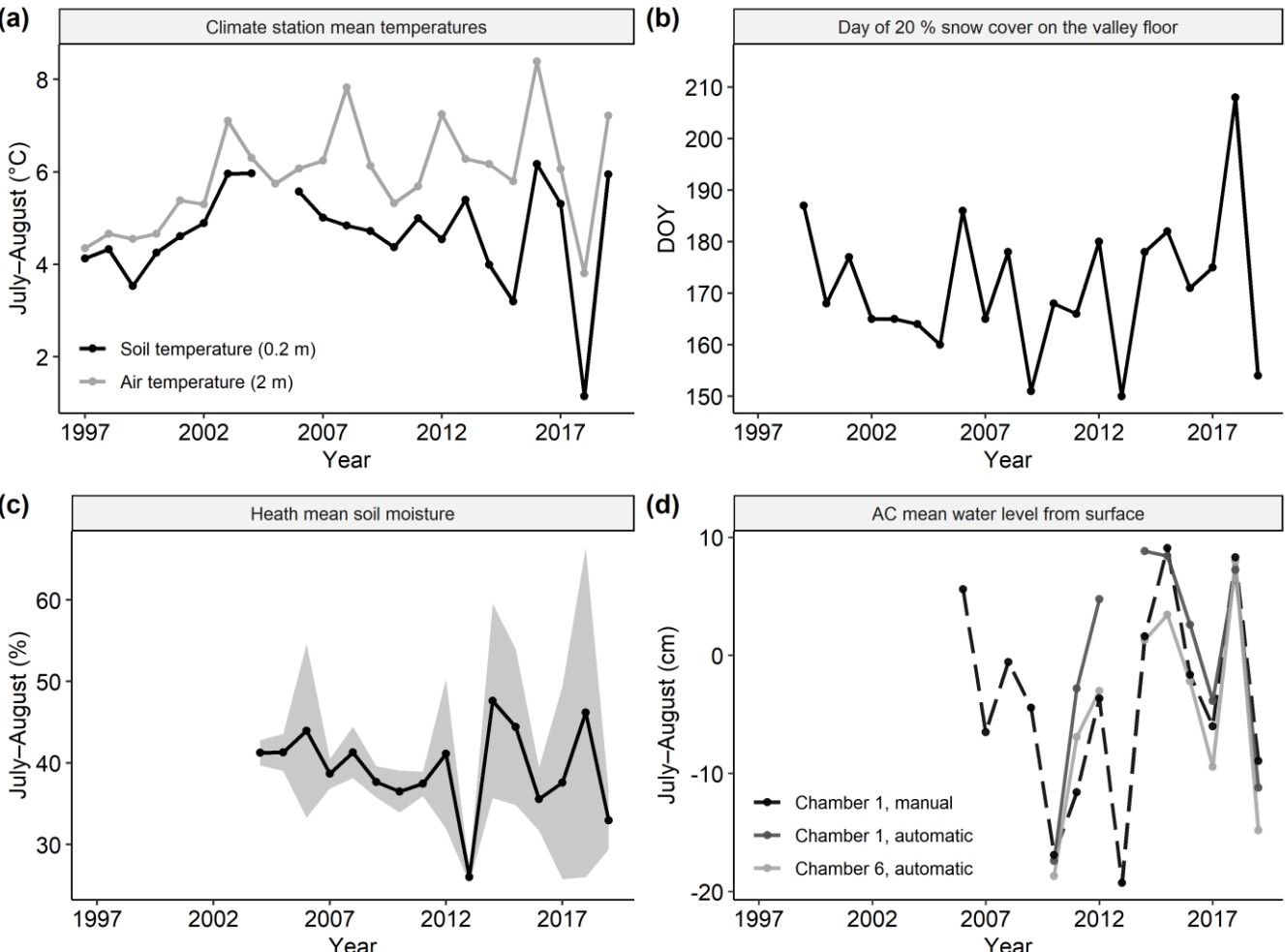

**Figure 3. (a) July–August mean temperatures measured at the Zackenberg climate station. (b) Timing of 20 % remaining snow cover on the valley floor with 1999–2014 data from Pedersen et al. (2016). (c) July–August mean soil moisture at the Mix-1 heath site in the lower valley with standard deviation (SD) as shading. (d) July–August mean water level relative to the surface measured close to chamber 1 and chamber 6 at the AC site.** *Data sources: GEM ClimateBasis and GeoBasis Zackenberg.*

### 3.2 Timeline of methane fluxes from Rylekærene

Figure 4 summarizes the methane fluxes measured on fen and fen fringe (only AC 1 to 6) surface types in Zackenberg Valley during 17 growing seasons from 1997–2019. The methane fluxes vary both between simultaneous measurements and between years. Fen fluxes generally show higher fluxes than those measured at the fen fringe, with mean fluxes ranging from $1.75 \pm 0.27$ to $8.3 \pm 0.31$ mg m$^{-2}$ h$^{-1}$ in 1997 (Friborg et al., 2000) and 2000 (Joabsson and Christensen, 2001). Methane fluxes in the fen fringe ranged from $0.26 \pm 0.07$ mg m$^{-2}$ h$^{-1}$ in 2011 to $3.41 \pm 0.57$ mg m$^{-2}$ h$^{-1}$ in 2007 during July–August (Mastepanov et al., 2013; Mastepanov et al., in prep.).

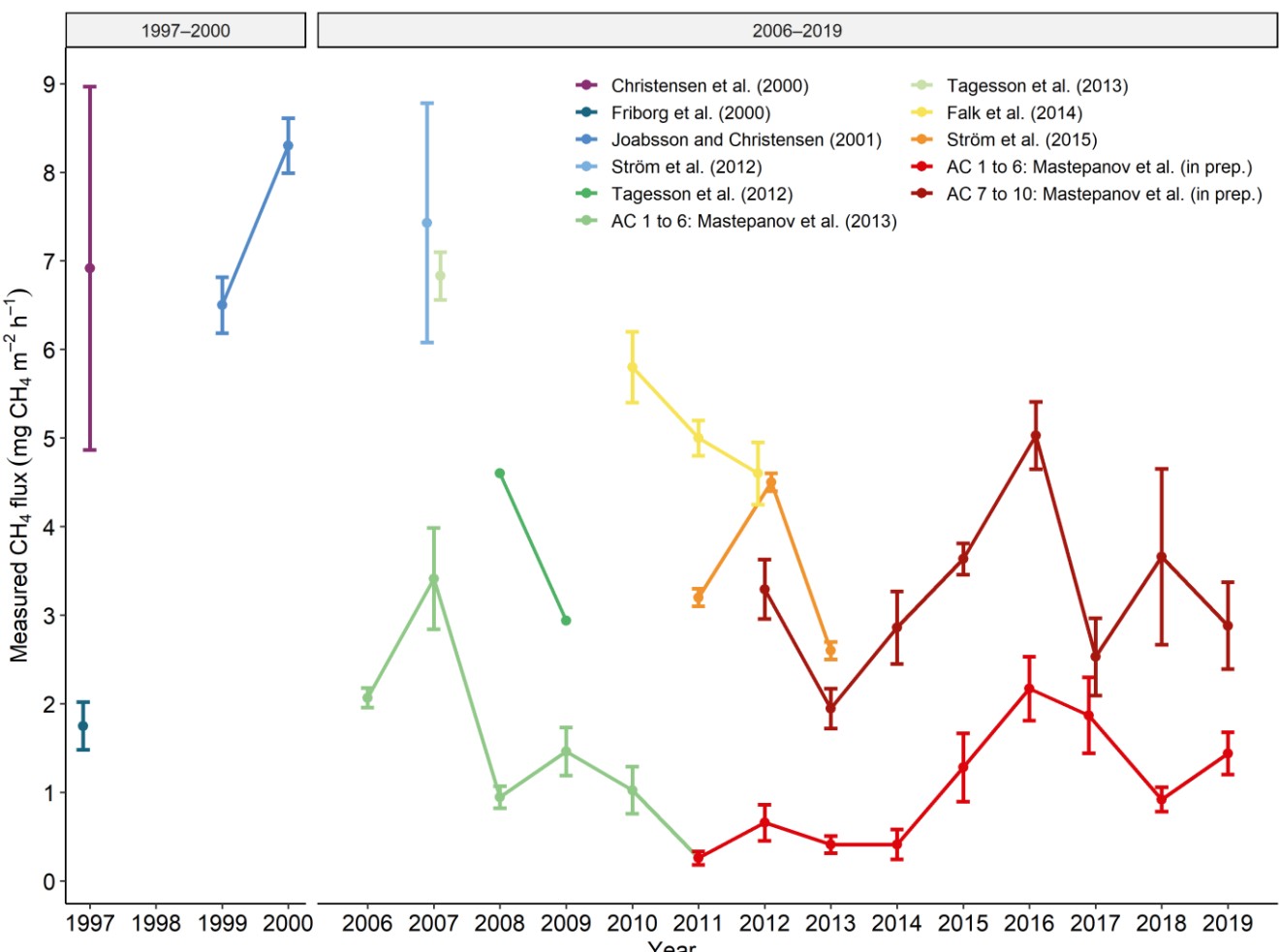

Figure 4. Timeline of methane fluxes measured during the growing season (1 July–31 August for AC) measured in fen areas (hummocky and continuous fen) and at the fringe of the fen (AC 1 to 6). Colors show different datasets. Please note that the years 2001–2005 are left out, as no published studies focusing on methane fluxes in undisturbed areas were made in this period. Also, note that points may be shifted slightly along the year axis to avoid overplotting.

### 3.2 Gully methane fluxes

Methane fluxes at exposed, eroded surfaces of the gully were different from the fluxes on the nearby, undisturbed surfaces (Fig. 5). The late growing season mean methane flux of the recently eroded surfaces in the gully in 2019 was $0.05 \pm 0.02$ mg m$^{-2}$ h$^{-1}$, including positive and negative fluxes. The grassland surface cover in areas not disturbed by erosion shows a negative methane flux of $-0.06 \pm 0.01$ mg m$^{-2}$ h$^{-1}$. The mean methane flux in the fen was $3.83 \pm 0.76$ mg m$^{-2}$ h$^{-1}$, more than 75 times higher than the mean flux in the gully. For several plots in the fen, the flux was highly variable over time, reaching 20 mg m$^{-2}$ h$^{-1}$. Generally, the methane flux decreased with air temperature in the fen. In the gully, plots with patches of live vegetation showed mostly negative flux, while barren plots exhibited a positive flux. In the undisturbed grassland areas, vegetated and barren plot fluxes were negative. Methane emissions generally increased closer to the open water body of the nearby fen. The emergence of the gully changed the area from a small methane sink to a small source, following an initial substantial episodic release of methane stored in exposed ice in 2018 when the gully appeared (Christensen et al., 2020b).

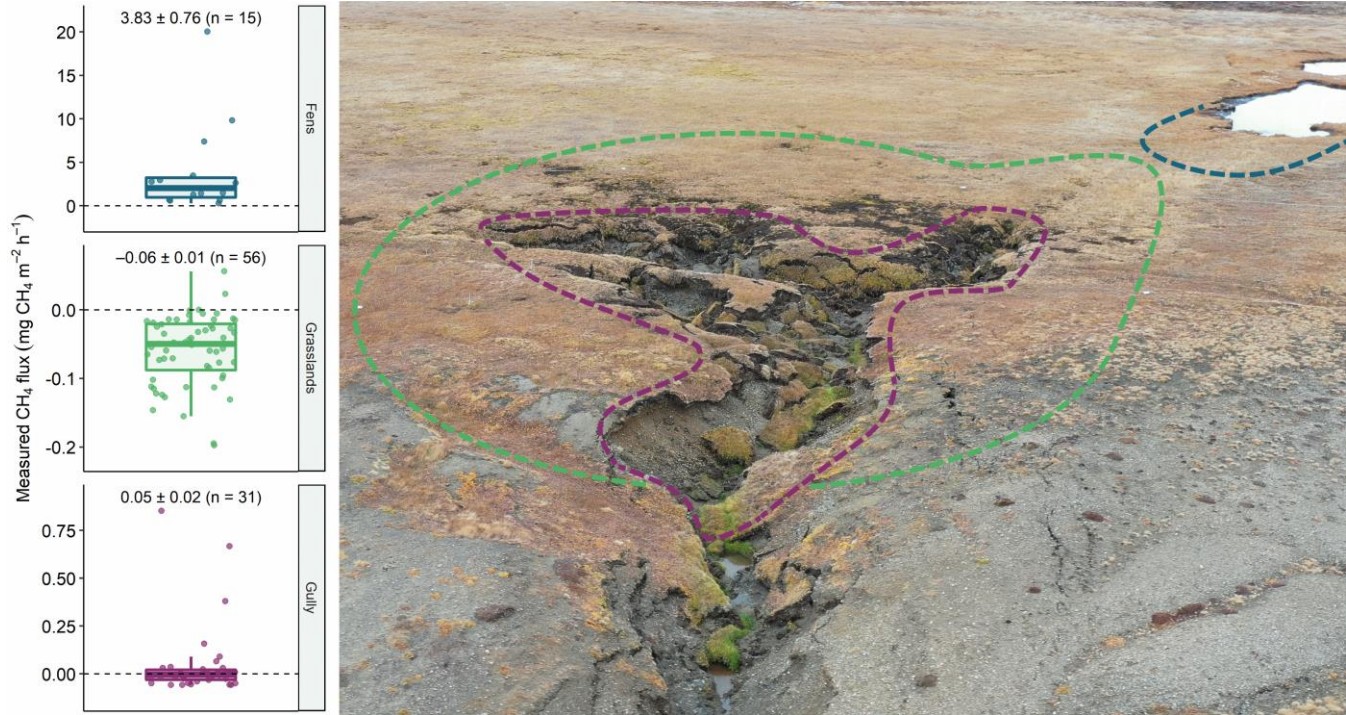

**Figure 5. The gully and fen area near the Zackenberg Research Station. The three boxplots show the measured methane fluxes on different surface types between 23 August and 1 September 2019, when the image was captured. Dots on top of the boxplots show all the individual measurements. The dashed lines in the image indicate the approximate spatial extent of the measurements near the gully. The eroded area is ~50 m long and ~25 m at its widest, covering ~720 m².**

### 3.3 Estimation of an integrated flux of methane in Zackenberg Valley

The July–August landscape flux of methane exposed large interannual variability over the valley floor and Rylekærene study areas over the 14 years, 2006–2019 (Fig. 6). The cumulative landscape fluxes and uncertainties are displayed as line plots and error bars overlapping stacked bar charts, which show the contribution from each of the six surface classes. Surface types with negative methane flux lower the landscape flux and SE for each of the surface classes on the stacked bar charts.

We observed no apparent trends for both areas ($n = 14$, *Pearson's r* $= -0.22$, $p = 0.44$) over the entire period. The mean flux for the 2006–2019 period in Rylekærene was $1.74 \pm 0.16$ mg m$^{-2}$ h$^{-1}$ and $0.34 \pm 0.03$ mg m$^{-2}$ h$^{-1}$ for the valley floor study area. Methane fluxes ranged from $0.98 \pm 0.11$ mg m$^{-2}$ h$^{-1}$ in 2011 to $3.26 \pm 1.15$ mg m$^{-2}$ h$^{-1}$ in 2007 in Rylekærene and from $0.17 \pm 0.05$ mg m$^{-2}$ h$^{-1}$ to $0.67 \pm 0.23$ mg m$^{-2}$ h$^{-1}$ for the valley floor. The two study areas were net sources of methane throughout the period, although its magnitude changed significantly between years.

Areas with a high positive flux are, in relative terms, more dominant in Rylekærene than the valley floor, which instead has a more considerable negative flux from heaths, *Salix* snowbeds, and fell and barren areas, which decrease the mean landscape flux.

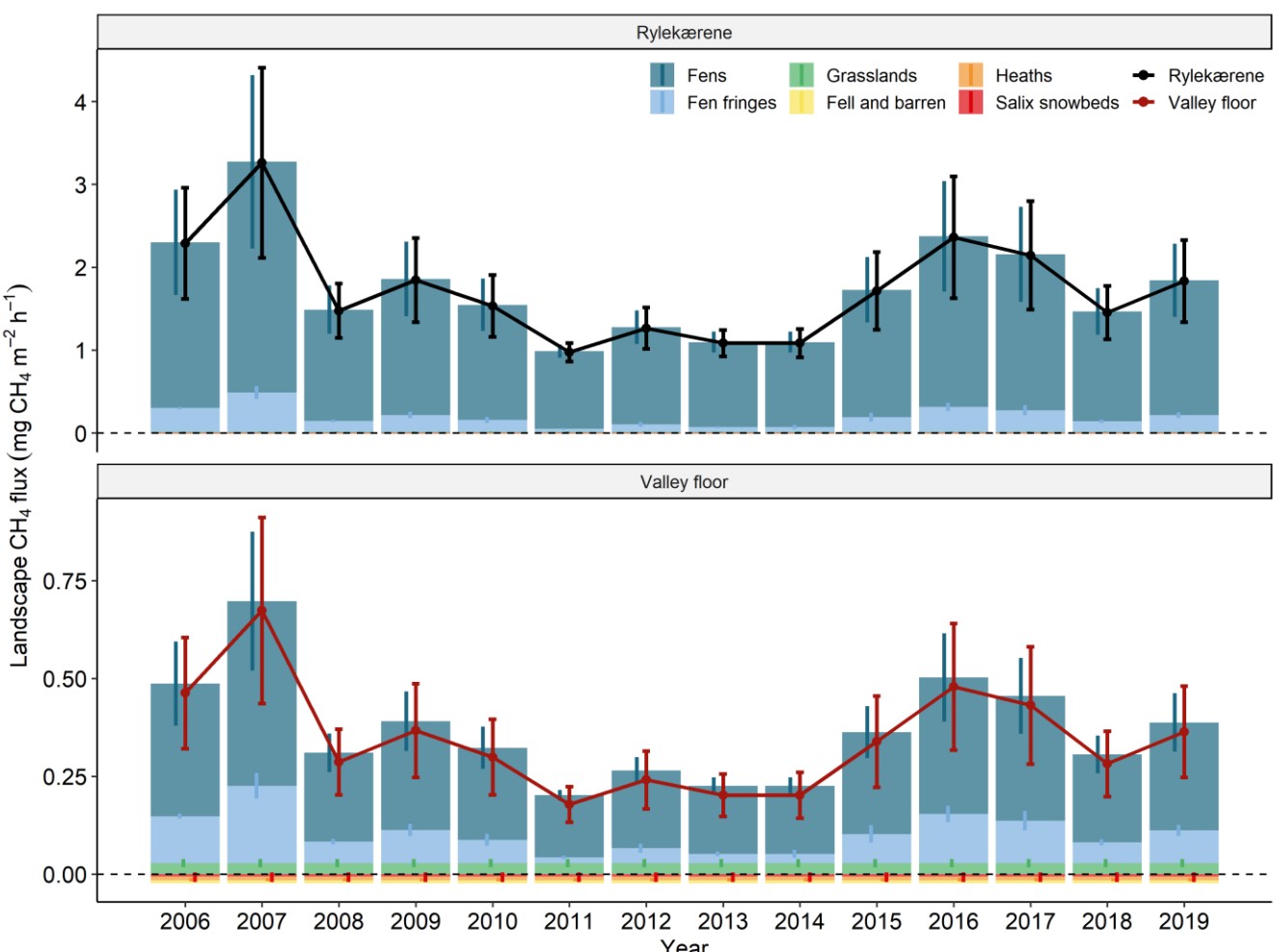

**Figure 6. Cumulative fluxes and SE from the six surface classes are shown for the two areas. Fluxes and SE from each surface class are weighted to their relative coverage and shown with stacked bars. The fen and fen fringe fluxes are predominant in the calculation of both landscape fluxes, explaining the nearly identical pattern in interannual variability.**

### 3.5 Methane emissions from the valley in a changing climate

The mean valley methane flux for July–August is calculated for 2016–2100, including the upper and lower bounds for the fluxes, when uncertainties from both the methane-temperature regression and the upscaling are combined (Fig. 7). For this comparison, mean fluxes are averaged over two periods, one for the mid-21$^{st}$ century (2041–2060) and another for the late-21$^{st}$ century (2081–2100). The mean landscape flux in the reference period (2008–2015) was 0.27 mg m$^{-2}$ h$^{-1}$ (Fig. 7, a), while the mean flux from 2016–2019 (Fig. 7, b) was 0.39 mg m$^{-2}$ h$^{-1}$. The modeled landscape flux without erosion (Fig. 7, c) increases linearly over the century. The modeled flux with erosion rates of 720 m$^2$ y$^{-1}$ (Fig. 7, d), 720–3600 m$^2$ y$^{-1}$ (Fig. 7, e), and 720–7200 m$^2$ y$^{-1}$ (Fig. 7, f) all result in a net reduction in landscape flux, which becomes more pronounced over time. Higher rates of erosion result in lower modeled flux for Zackenberg Valley.

When considering the differences between the fluxes shown in Fig. 7, the mid-century mean landscape flux could be reduced by up to 0.05 mg m$^{-2}$ h$^{-1}$, i.e., the difference between (Fig. 7, c) and (Fig. 7, f) corresponding to a

375 reduction of 1.2 % of the landscape flux. Between 2081–2100, the reduction in methane flux for the most severe erosion pathway could be 0.029 mg m$^{-2}$ h$^{-1}$, equal to a reduction in the landscape flux of 4.9 %.

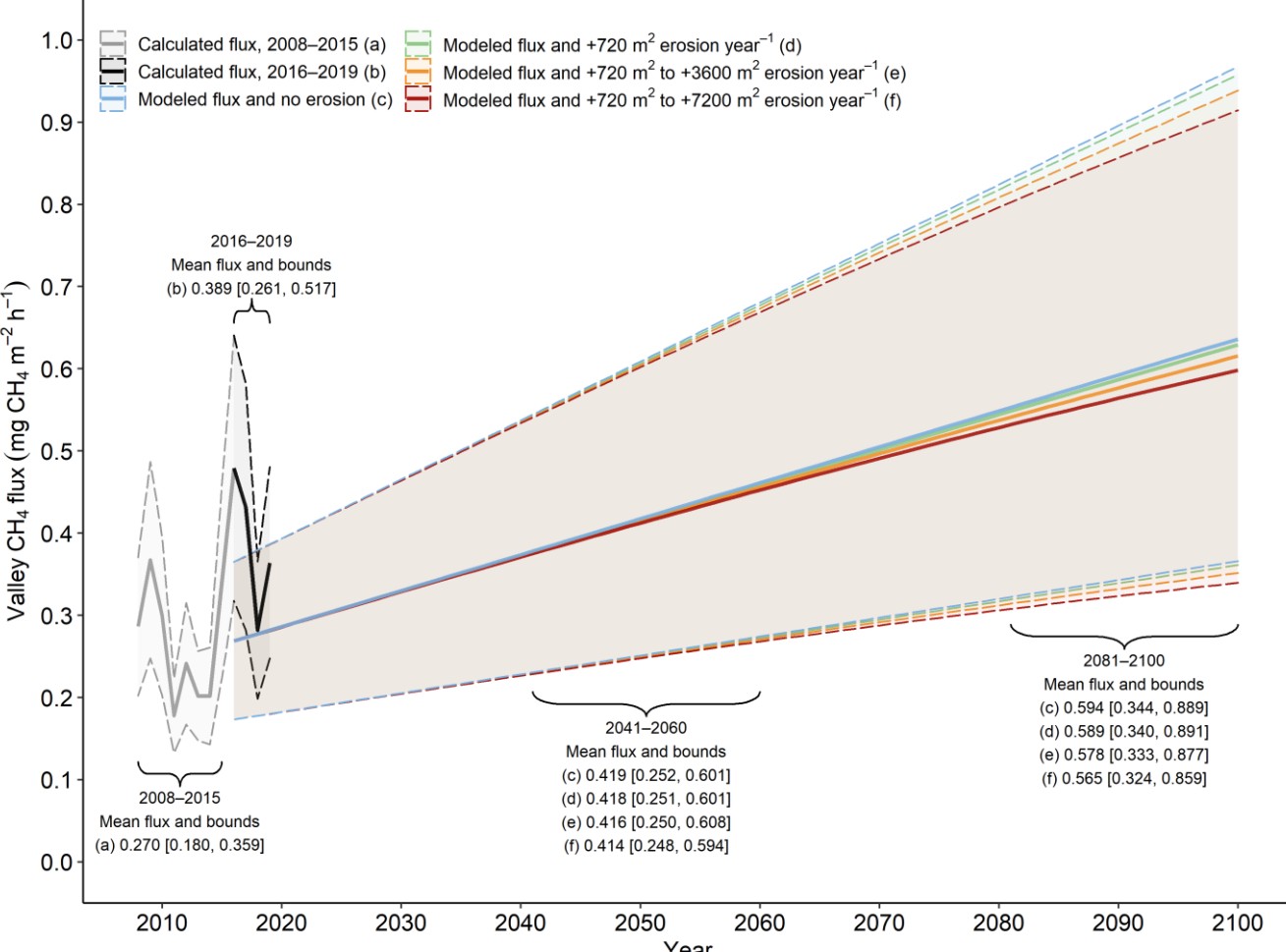

**Figure 7. Methane fluxes in the valley floor study area increase over time if temperatures rise as modeled for the RCP8.5 scenario. However, increases in flux from rising temperatures are partly offset by erosion in the sensitivity study. Increasing erosion rates**
**could reduce the mean flux of methane in the valley over time when a part of the landscape, a net source of methane, is converted into eroded areas with lower methane flux.**

## 4 Discussion

### 4.1 Methane flux measurements

Several methane flux methods have been in use since the two first methane flux studies were carried out in 1997
(Fig. 4) in the Rylekærene area (Christensen et al., 2000; Friborg et al., 2000), including both manual chambers (Table 2) and EC measurements. Flux measurements over 14 years at AC show large interannual variability in methane fluxes even at a single site. Substantial spatial variability in fluxes is seen in Fig. 4 when studies are available in the same years. Differences in methane fluxes measured from these studies can be explained by site characteristics, equipment and sampling strategies, and both length and timing of the measurement period. Two
studies are omitted in Fig. 4: Pirk et al. (2016b) studied the methane fluxes in the fen areas near the AC site under

snow-covered conditions before the growing season in 2012 and 2014. Jørgensen et al. (2015) added flux measurements from 2012 covering only 'dry tundra' (*Dryas* heath, abrasion plateau, and fell field) and 'moist tundra' (*Salix* snowbeds and *Cassiope* heath) sites, both of them methane sinks. These findings exceeded the methane uptake on similar tundra surface classes in the area in Christensen et al. (2000). Higher soil and air temperatures in 2012 compared to 1997 could explain the higher fluxes (Fig. 3a), which also fits with relatively high methane uptake in 2007 when Tagesson et al. (2013) did their measurements on corresponding vegetation types. Spatial differences and the inclusion of unvegetated surfaces (abrasion plateau and fell field) and more advanced measurement equipment (Table 2) are other possible reasons for the larger uptake in Jørgensen et al. (2015).

Using a distributed manual chamber approach, Christensen et al. (2000) measured fluxes for an intensive study area, a part of Rylekærene, in 1997. In Fig. 4, only the combined measurements from hummocky and continuous fen plots are used. Alongside, Friborg et al. (2000) applied the EC method and found a much lower methane flux compared to Christensen et al. (2000) in approximately the same area. The fetch of the EC tower included surface classes with lower flux, which is a likely reason for lower mean flux Rylekærene reported in Friborg et al. (2000). The length of the measurement period was longer in Friborg et al. (2000), starting 1 June (Fig. 2), more than three weeks earlier than Christensen et al. (2000), which further explains some of the differences as methane fluxes were low at the beginning of the measurement period. Christensen et al. (2000) measured methane fluxes two times per week per plot, while methane fluxes from EC were available every day for several weeks in July and August. The SE for the fen measurements in Christensen et al. (2000) in Fig. 4 is high due to the high flux variability within and between the two combined groups, hummocky and continuous fen.

Control plot flux measurements from Joabsson and Christensen (2001) are illustrated in Fig. 4 for a study site in the southern part of Rylekærene in 1999 and 2000. As shown in Table 2, the methods are comparable to Christensen et al. (2000), and the mean flux in both years is similar. The SE are smaller, likely due to both fewer and less diverse plots, i.e., six continuous plots compared to a total of 18 hummocky and continuous plots in Christensen et al. (2000).

Two studies provide a basis for comparison in 2007, the growing season with the highest estimated landscape flux in the 14-year study period. Ström et al. (2012) and Tagesson et al. (2013) applied similar methods (Table 2), using plots near MM2 in the same year with comparable methane flux as a result. For Ström et al. (2012), the SE reported in Fig. 4 is comparatively high. This difference can be explained by the high SE associated with measurements of plots with high flux and high *Eriophorum scheuchzeri* coverage, combined with relatively low methane flux plots with low *Eriophorum scheuchzeri* coverage. In Tagesson et al. (2013), measurements were done at 25 randomly-placed plots in hummocky and continuous fen areas, which resulted in a lower SE for the combined fen classes. The key difference is the range of both high and low *Eriophorum scheuchzeri* plots in Ström et al. (2012) against the randomly selected placement in two fen classes in Tagesson et al. (2013).

In 2008 and 2009, Tagesson et al. (2012) combined gradient and EC methods to estimate methane fluxes. Their results for the two growing seasons show intermediate fluxes between the fluxes measured at the fen fringe and higher fluxes measured by chamber methods on fen surfaces. The size of the fluxes can be explained by the limited 43 % coverage of the fen surface class in the tower footprint for the prevailing wind direction in the growing season, while the rest of the footprint is covered by drier surface classes. Error estimates are not directly comparable with the other studies in Fig. 4. They are integrated over the entire measurement period and reported as total accumulated errors equal to 8 % and 12 % of the total flux in 2008 and 2009.

Based on control measurements in a continuous fen area during the growing seasons 2010–2012, Falk et al. (2014) reported mean methane fluxes larger than those measured at the fen fringe at the AC. Measurement methods are comparable to Ström et al. (2012) and Tagesson et al. (2013). They use an identical gas analyzer and similar measurement frequency. Different chamber sizes are used, but this is not expected to impact the measurements significantly, as chamber volumes are considered in the calculation of fluxes. Differences in mean flux between the three years were attributed to variable measurement periods in the three years and interannual variations in water level depth (Falk et al., 2014).

Ström et al. (2015) measured methane fluxes with the same approach as Falk et al. (2014). The main differences between the two studies are the timing and the site. The two sites are situated ~600 meters apart, which explains possible differences in fluxes due to water levels and substrate availability. Measurements in 2011 in Falk et al. (2014) began 14 days earlier, which may have caused some of the difference in mean fluxes. In 2012, fluxes were similar between the two studies. The two campaigns started one day apart in that year, indicating that conditions are similar despite the distance.

The outer four chambers of the AC installation show interannual variability in fluxes of a size similar to the manual chamber measurements of Falk et al. (2014), Ström et al. (2015), and EC measurements in Tagesson et al. (2012). The number of observations for chambers 7 to 10 is high (Table 3), and flux measurements from each chamber are available every 90 minutes during July and August. In some years, measurements are not available for the entire period. Differences in chamber fluxes may be explained by variability in water levels, but data not available from the outermost chambers. Standard errors are higher due to relatively high variability in fluxes during the measurement period.

The previous studies show large differences in methane flux within the same fen ecosystem. Local spatial variability, interannual differences may explain these differences, and possibly also by differences in methods. Spatial variability in methane flux can be pronounced in tundra ecosystems with complex microtopography (Olefeldt et al., 2013) and differ significantly even at a meter-scale (Fig. 4). Methane fluxes vary with local differences in substrate, water-table depth, grazing, vegetation composition and productivity, each of which can

either increase or limit the methane flux. Several of these interactions have been studied in Zackenberg Valley: increased substrate availability, mainly acetate, contributes to higher methane fluxes as shown by, e.g., Ström et al. (2003), while a low water table limits methane fluxes (Tagesson et al., 2013). Grazing can either decrease (Falk et al., 2014) or increase (Falk et al., 2015) the methane flux, depending on the vegetation cover. Vegetation composition and primary productivity are strong drivers of methane fluxes (Joabsson and Christensen, 2001; Ström et al., 2012; Ström et al., 2015).

Temporal variability in methane fluxes can be caused by differences in environmental parameters, both within a growing season and from one year to another. Soil temperature was found to explain less variability than species composition and primary productivity (Christensen et al., 2000; Ström et al., 2012), while soil temperatures showed a high correlation with methane flux within most individual years (Mastepanov et al., 2013). Late-lying snow delays the beginning of the growing season (Grøndahl et al., 2008), which controls several of the above-mentioned parameters. Only a few studies in Zackenberg Valley were conducted outside the growing season. However, the fall season could substantially impact the annual methane budget, mainly through emissions associated with the onset of soil freezing (Mastepanov et al., 2008). In contrast, wintertime (November–May) emissions may only have a limited impact on the annual methane budget (Pirk et al., 2016b).

Our analysis focuses on the mean of the July–August fluxes in the valley, but the automated chambers are running from snowmelt to the end of the field season (Fig. 2). The fixed period matches the timing of the previous studies, and the period showed a good representation of the mean flux of the entire measurement dataset. The first 30 to 40 days after snowmelt have been shown to express the main differences between years (Mastepanov et al., 2013), which is covered by the July–August mean fluxes to a large extent.

## 4.2 Combining flux measurements from multiple sources

Comparable mean methane fluxes across several existing studies indicate that the differences in methods and placement have an impact on the flux in fen areas. Differences arise mainly when measurement periods vary by several weeks and when measurement plots are distributed over several different surface classes, i.e., hummocky and continuous fen. However, the fen fringe time series appears to express some of the variability in fluxes as the separate measurement campaigns when these are combined into one. Regression analysis shows a significant correlation between methane fluxes at chambers 1 to 6 and the grouped studies from the fen areas of Zackenberg Valley when pairing data for the same years. Deming regression is a reasonable choice of analysis when both the mean values of chamber 1 to 6 and the fen methane fluxes are associated with measurement errors and uncertainties. The composite fen dataset, made from six separate studies, increases the temporal coverage and the spatial coverage of the data used in the regression analysis. The SE of the calculated fluxes is estimated with jackknife resampling, providing a reasonable SE estimate for Deming regression (Linnet, 1990). When comparing flux data to OLS regression, the Deming regression slope is steeper. The steeper slope means that the estimated fen flux in 2007, when the flux was high, is ~20 % higher when compared to OLS, which means that the difference in regression methods has a substantial impact on landscape fluxes.

Methane fluxes measured in chambers 1 to 6 are consistently lower than the fluxes in chambers 7 to 10. Therefore, data from chambers 1 to 6 are split into a separate surface class, the fen fringe. The addition of these lower flux fen areas impacts the landscape flux, but the width of the zone does most likely fluctuate across the valley. Here, 10 m is a simple estimate based on the situation at AC, but the width of the zone ultimately depends on the topographic gradient along the edge of the fens.

The remaining surface classes contribute with the same negative fluxes throughout the period, with a mean value representing each of the surface classes. Their net effect is offsetting some of the positive landscape methane flux, particularly for the valley floor. Several surface classes were studied in detail over a single growing season (Table 3), and the specific sites differ between studies. The fluxes from these areas are held constant in lack of more detailed data, and fluxes from different studies are averaged into a single mean flux, which represents a range of sites and environmental conditions. We omitted surface flux data from the grasslands, measured by Christensen et al. (2000). The measured mean flux was significantly higher at $2.1 \pm 1.6$ mg m$^{-2}$ h$^{-1}$, and the measurement site was located near the border of the fen, see Fig. 2 in Christensen et al. (2000), making the data more comparable to the fen fringe than the grasslands surface class.

Manual measurements in the gully area were limited to 10 days in the late growing season. Still, the size of the mean fluxes is in good agreement with the concurrent fluxes measured at AC 7 to 10 (Mastepanov et al., in prep.) and the grassland flux reported in Tagesson et al. (2013). Negative fluxes were measured on plots with patchy vegetation in the gully, while mineral-rich plots had low positive fluxes of methane. The highest methane emissions in the gully were found on two unvegetated plots, located where the gully had recently eroded (up to 0.85 mg m$^{-2}$ h$^{-1}$). These sparse observations suggest a considerable reduction in flux on eroded gully surfaces over time when limited organic soils remain in the gully.

### 4.3 Methane flux upscaling

Vegetation plot measurement on the dominant vegetation classes combined with long-term AC site data from 2006–2019 enables the calculation of landscape flux time series for Rylekærene and the valley floor. This approach is more direct than modeling based on physical parameters that have otherwise been shown to correlate well with methane flux for Rylekærene (Tagesson et al., 2013). However, a limitation of the approach is its lack of a dynamic component, which could take spatial differences across the valley into account, e.g., snow cover or changing soil moisture conditions from one year to another, affecting the valley floor and the AC site differently.

Both Christensen et al. (2000) and Tagesson et al. (2013) found a significant difference in the methane flux in hummocky and continuous fen areas in Rylekærene and treated the two groups separately. These groups were combined in this study to match the existing valley-wide surface cover classification. Comparing the mean flux found for Rylekærene shows good agreement between the flux estimates in this study and the fluxes reported in Tagesson et al. (2013). Four growing seasons (2007–2010) are common for the two studies, and in three of the seasons, the flux estimates of this study lie within the model uncertainty in Tagesson et al. (2013). The one notable exemption is 2007 mean flux ($3.26 \pm 1.15$ mg m$^{-2}$ h$^{-1}$) in this study, and it differs a lot from the $1.6 \pm 1.0$ mg m$^{-2}$

h$^{-1}$ modeling result in Tagesson et al. (2013). The difference can be explained with a relatively high flux measured at the AC site in a year without extreme temperature or moisture conditions in the valley (Fig. 2), which are central parameters in the modeling of Tagesson et al. (2013).

The AC site is located by the outlet of the Rylekærene fen. The flow of water through the area affects the water level at the site, which is shown in Fig. 3d. Water level data was measured manually, once per day in the growing season, between 2006–2019 near the outermost of the six original chambers (chamber 1). Automatic water level measurements are available for chamber 1 and chamber 6, the innermost chamber, in the 2010–2019 period, with

a gap in data for 2013. For July–August, water levels are generally lower at the innermost chamber but show variability similar to the water levels measured further out in the fen. Water level measurements are not available for the outer four chambers.

The changing water level may control the methane flux and explain its high variability 2006–2019, but neither water levels, air temperatures, nor soil temperatures correlate significantly (*p*-value threshold of 0.05) with methane

when analyzing the interannual variability for July–August mean values. Over the 14 years, *p*-values range from 0.29 to 0.56. The lack of correlation over the time series illustrates a complex interaction between methane and environmental conditions when analyzed on a decadal scale. Figure 4 shows the methane flux of previous studies, which mainly focused on wetter parts of the fen where the flux is high. The flux appears to vary relatively less over time further out in the fen, as the soil moisture conditions change less between years than the chambers 1 to 6 at

the AC site. One example is the smaller relative variability at chambers 7 to 10. Similar, large variability could common for the smaller, discontinuous fen areas in the valley seen in Fig. 2. These areas are characterized by a less stable inflow of water than Rylekærene, which may therefore cause high interannual variability in the transition zones between different vegetation types.

Christensen et al. (2000) based their study of Zackenberg Valley methane flux on chamber measurements from a ~0.1 km$^2$ area of the northern part of Rylekærene (Fig. 1). The methane flux was measured with chambers on the five dominant vegetation types, and a mean flux for the valley was calculated by scaling fluxes to match the land cover classification of Bay (1998). The upscaled methane flux for the entire valley floor was $1.9 \pm 0.7$ mg m$^{-2}$ h$^{-1}$, which is higher than in any of the years in this study, which range from $0.18 \pm 0.05$ mg m$^{-2}$ h$^{-1}$ and $0.67 \pm 0.24$ mg

m$^{-2}$ h$^{-1}$. The distribution of land cover classes differs slightly between Bay (1998) and the HyMap dataset. Still, a primary cause for the much higher valley estimate is the higher measured fluxes in the widespread grassland class with a flux of $2.9 \pm 1.6$ mg m$^{-2}$ h$^{-1}$ in Christensen et al. (2000), compared to $0.1 \pm 0.04$ mg m$^{-2}$ h$^{-1}$ (SE converted from SD) in Tagesson et al. (2013), the source of the grassland flux data used in our study. This difference impacts the valley flux, and if we substitute only the grassland-type flux from 0.1 mg m$^{-2}$ h$^{-1}$ to 2.9 mg m$^{-2}$ h$^{-1}$ in the

upscaling to the entire valley floor, it would at least double the methane flux in the valley.

Jørgensen et al. (2015) found a relatively high methane uptake on dry tundra in 2012, e.g., *Salix* snowbeds and Heath surfaces, compared to what has been found earlier. When including the uptake from Jørgensen et al. (2015), the valley flux is reduced, but the effect could vary between years with larger sinks in dry, warm years, as these

surface classes combined account for more than 40 % of the valley floor (Table 1). Soil temperatures were lower in 2012 than in 2007 in July-August, which does not explain the higher uptake on dry tundra soils in 2012, indicating that some dry tundra surfaces have a higher methane uptake than others. Additionally, flux data from 2007 are used for both heath, grassland, and *Salix* snowbeds surface classes. While flux measurements in the fen areas were exceptionally high in that year, this was not the case for the drier surface types. Landscape fluxes from both the valley floor and Rylekærene are highly variable between years and do not show an increasing trend from the available data. Both study areas show the same pattern, but the areal distribution between classes differs, making the Rylekærene study area almost entirely dependent on the variability of only the fen and fen fringe surface types. In the larger valley floor study area, grassland, *Salix* snowbeds, heaths, and fell and barren surface classes dampen the variability between years.

Zackenberg Valley shows the potential for increased methane fluxes in the 21st century, as methane shows a positive correlation with temperatures (Geng et al., 2019), similar to the rest of Greenland and the Arctic (AMAP, 2017).

Landscape-scale methane flux estimations are available from other subarctic and tundra sites (Table 4) in North America, Scandinavia, and Russia. The upscaled growing season fluxes range from 0.5 km$^2$, extending from covering a few different ecosystems in a single site (Christensen et al., 2004) to large-scale estimates of fluxes covering up to 320,000 km$^2$ for the Hudson Bay Lowlands (Roulet et al., 1994). Mean methane fluxes in these landscapes range from 0.3 to 3 mg m$^{-2}$ h$^{-1}$ across different scales and landscape types, covering growing seasons of variable length with studies scattered across several decades. The mean methane flux for Zackenberg Valley from 2006–2019 is 0.34 mg m$^{-2}$ h$^{-1}$ in the valley floor area (~16 km$^2$) and 1.74 mg m$^{-2}$ h$^{-1}$ in the fen-rich Rylekærene area (~1.3 km$^2$). Hence, the results from Zackenberg Valley are in good agreement with observations from comparable studies at other sites. However, the two Zackenberg study areas are smaller than most studies listed in Table 4. All the included studies are either fully or partly based on chamber measurements and upscaling with areal coverage of the surface classes, making them comparable to this study. Several studies (Bartlett et al., 1992; Roulet et al., 1994; Bosse and Frenzel, 2001; Hartley et al., 2015) estimated the landscape methane fluxes based on observations from a single year. The remaining studies combine observations from multiple years or studies to a flux estimate from each surface cover class (Christensen et al., 2004; Schneider et al., 2009; Andresen et al., 2017; Morozumi et al., 2019). Flux measurements from several years lead to more robust landscape flux estimates, as the fluxes are highly variable between years.

The large differences in study area size and composition ultimately determine the mean methane flux estimates of the landscape, making direct comparisons between sites difficult. For instance, the mean landscape flux found in this study is nearly five times greater for the fen-rich Rylekærene study area, which is fully contained in the valley floor study area. On an even larger scale, the entire northeast Greenland acts as a net sink of methane, as Jørgensen et al. (2015) found a mean methane flux of ~−0.08 mg m$^{-2}$ h$^{-1}$ in their 10,675 km$^2$ study area.

**Table 4. Comparison of landscape-integrated growing season methane flux for various subarctic and Arctic sites with a minimum size of 0.5 km².**

| Publication | Location | Climate zone | Landscape type | Mean flux (mg CH$_4$ m$^{-2}$ h$^{-1}$) | Area size (km²) |
|---|---|---|---|---|---|
| Bartlett et al. (1992) | Yukon-Kuskokwim Delta (Alaska, USA) | Subarctic | Wetlands | 1.8 | 97,400 |
| Roulet et al. (1994) | Hudson Bay Lowlands (Ontario-Manitoba, Canada) | Subarctic | Wetlands | 0.8 | 320,000 |
| Bosse and Frenzel (2001) | Yenisey River (W Siberia, Russia) | Subarctic | Mire, wetlands, and Pine forest | 1 | 361 |
| Christensen et al. (2004) | Stordalen (Norrbotten, Sweden) | Subarctic | Mire | 2.7 to 3.0 | 0.5 |
| Heikkinen et al. (2004) | Lek Vorkuta (N Komi, Russia) | Arctic tundra | Heath, peatland, and Salix | 0.6 | 114 |
| Schneider et al. (2009) | Lena Delta (N Siberia, Russia) | Arctic tundra | Wetlands | 0.4 | 29,036 |
| Hartley et al. (2015) | Kevo (Lapland, Finland) | Subarctic | Aapa mires and birch forest | 0.3 to 0.4 | 100 |
| Andresen et al. (2017) | Utqiaġvik Peninsula (Alaska, USA) | Arctic tundra | Arctic coastal plains | 0.6 | 1779 |
| Morozumi et al. (2019) | Indigirka (NE Siberia, Russia) | Arctic tundra | Larch forest, shrubs, and wetlands | 1.6 | 96 |

Several other studies have applied the EC method for ecosystem methane flux measurements on a landscape scale in the Arctic, e.g., Fan et al. (1992); Sachs et al. (2008); Wille et al. (2008); Jackowicz-Korczynski et al. (2010); Parmentier et al. (2011); Taylor et al. (2018). While the EC method requires less workload and integrates ecosystem fluxes at high temporal resolution by nonintrusive means (McGuire et al., 2012), those studies are generally restricted to smaller areas less than 0.5 km². Ecosystem fluxes from these studies range between ~0.1 to 6.2 mg m$^{-2}$ h$^{-1}$, with the lowest growing season fluxes measured at an upland tussock tundra site by Eight Mile Lake in Alaska (Taylor et al., 2018) and the highest fluxes measured in a mire in Stordalen, north Sweden (Jackowicz-Korczynski et al., 2010). Mean growing season fluxes found in Zackenberg using the EC method are within this range (Friborg et al., 2000; Tagesson et al., 2012).

## 4.5 Landscape methane flux in a changing climate

A warming trend in both air and soil temperatures has been observed for Zackenberg Valley (Abermann et al., 2017; Christensen et al., 2020b). The increase in temperatures has contributed to the destabilization of permafrost, leading to several active periglacial landforms in recent years (Docherty et al., 2017; Cable et al., 2018; Christensen et al., 2020b). Modeling results show higher soil temperatures and a deepening of the AL in Zackenberg in the future (Christiansen et al., 2008; Westermann et al., 2015). Increasing temperatures are expected to impact both positive and negative methane flux and surface erosion in the Arctic (Schuur et al., 2015; Geng et al., 2019; Oh et al., 2020), which is also likely for Zackenberg Valley. The emergence of several active erosion sites in Zackenberg Valley in recent years could be an initial step towards increased erosion activity during the 21st century. In 2019, after the disappearance of methane-rich ice wedges in the previous year, carbon-rich soils had been washed out from the gully, leaving a silt-organic mix with limited potential for methane emission in the area (Christensen et al., 2020b). Our study defines three erosion pathways to illustrate the sensitivity of methane flux to land cover changes on a valley scale. We hypothesize large-scale linear growth in eroded areas in parts of the valley that are likely to have a high ground ice content and are located near streams and rivers identified by Cable et al. (2018). These erosion areas are assumed to share the characteristics of those observed in the gully area in 2019. Increased gully erosion could transform large areas with surfaces with both methane emission and uptake into well-drained,

low emission eroded surfaces. These erosion pathways are unlikely but they are valuable as a sensitivity study illustrating the difference in importance of increasing temperatures relative to eroding surfaces.

Even large-scale erosion on the valley floor would have a limited impact on the mean methane flux, reducing it by up to 1.2 % on average between 2041–2060 for the most extreme erosion pathway (Fig. 7, f). The reduction becomes more pronounced (4.9 % reduction) between 2081–2100, as eroded areas would continue to erode at a faster pace. Disturbances of this magnitude are comparable to the size of the edge trimming during the GLOF in 2017 in the lower river section (Tomczyk et al., 2020), located inside the valley floor study area. The impact on
fluxes from one added gully, similar in size to the recent gully per year, is minimal (less than 1 % reduction) at the end of the 21$^{st}$ century (Fig. 7, d) when compared to the uncertainty range. In our calculation of landscape methane fluxes after erosion, eroded areas become revegetated at a rate of 2 % y$^{-1}$. This rate may change over time and from one location to another, dependent on species composition and soil conditions in the eroded areas. The change is assumed to be linear in this case but may accelerate over time, as plant communities established. In the calculation,
revegetated areas were considered to have the same methane flux as the grasslands in the gully area. Limited data from the measurements in the gully area shows that the undisturbed vegetated areas take up slightly more methane than the disturbed vegetation patches in the gully, but the 10-day measurement period did not allow for more detailed measurements on temporal dynamics in the area.

The resulting changes in flux following erosion, particularly for the less extreme pathways (Fig. 7, d and e), are minor relative to the considerable uncertainty in the general shift in methane for the valley following an increase in temperature. The mean methane flux is shown in Fig. 7 with a wide uncertainty range, partly caused by a lack of perfect fit between fluxes and temperatures and the uncertainties in the estimated landscape fluxes in Fig. 6. The variability between years in this study is substantial. The mean valley flux for 2016–2019 exceeds the confidence
bounds, which shows that this should be seen as a sensitivity study of changes occurring on decadal scales.

In this case, methane fluxes were measured in a gully, which becomes more drained with a loss of organic soils from the surface after an erosion event (Christensen et al., 2020b).

The gully described in this study has likely formed as a direct consequence of pronounced lateral erosion from the river, which steepened the gully, allowing for increased drainage of water and sediments. The increased drainage
exposed more and more of the ice wedges and frozen soils, which later thawed and flowed out through the steeper gully. Lateral erosion of a similar scale has not been recorded to occur along the smaller rivers and streams in Zackenberg, but several smaller erosion sites have been described in recent years (Docherty et al., 2017; Cable et al., 2018). The gully shares characteristics with thermokarst gullies, a common type of thermokarst erosion, including extent, depth, and shape (Jorgenson et al., 2008). Abrupt thaw, including both gullies and thermokarst
areas, can take different forms and affect the surface methane flux through disturbances in both vegetation and hydrology (Turetsky et al., 2020). Olefeldt et al. (2016) describe thermokarst landscapes, which include thermo-erosion gullies characterized by lateral movement of sediments, similar to the gully described in our study. Olefeldt et al. (2016) estimate that ~20 % of land areas in the northern permafrost zone are thermokarst landscapes, meaning

they are either currently characterized by soil settlement or erosion or prone to developing into thermokarst landforms in the future. Thermokarst landforms have diverse impacts on a landscape, dependent on their type (Kokelj and Jorgenson, 2013). They could form abruptly and rapidly responding to increasing temperatures (Farquharson et al., 2019; Lewkowicz and Way, 2019). Wickland et al. (2020) found a 42 % increase in growing season methane flux from 1949–2018 in a wet polygonal tundra after an increase in thermokarst erosion of ice wedges. Thermokarst lakes cover large areas across the Boreal zone and in the Arctic and have been reported to be substantial emitters of methane (Wik et al., 2016; Walter Anthony et al., 2018; Engram et al., 2020). The findings from thermokarst lakes contrast the results from this study, where erosion causes draining and loss of organic material (Fig. 5). Wik et al. (2016) found a mean flux, combining diffusion and ebullition, of ~3.7 mg m$^{-2}$ h$^{-1}$ from thermokarst lakes in the ice-free season. Walter Anthony et al. (2018) found an accelerating increase in methane fluxes from thermokarst lakes in their modeling of methane in the 21$^{st}$ for the circumpolar region, which shows that the impacts on methane in the event of erosion are diverse. It should be noted that there is a difference in the 'receiving end' of the gully formation between our study and these thermokarst lake studies as the drainage in the Zackenberg valley goes into the river systems with few or no stagnant reservoirs between.

**5 Conclusions**

In this study, we have summarized 14 measurement-years of methane fluxes and several short-term campaigns, which provide a unique insight into the large variability in methane fluxes in a high Arctic tundra landscape of Zackenberg Valley. We have combined July–August measurements from a monitoring site running from 2006–2019 with detailed measurements of the most common vegetation types in the valley to estimate valley-wide methane fluxes over the period. For the valley, the net emission of methane in July–August shows differences by nearly a factor of 4 between individual years (2006–2019). Consistently dry or wet surfaces may remain relatively stable in terms of methane fluxes over the period, as indicated by the data from previous site-specific campaigns in the valley. However, the large areas covering the boundary between these hosts highly variable methane fluxes, significantly impacting methane fluxes on a landscape scale. Future multi-year campaigns should focus on measuring the full gradient from wet fens to dry heath to improve estimations of landscape methane fluxes, as the fluxes from different surface classes may respond differently to changes in environmental conditions, such as moisture, temperature, snow cover.

Observations from recently eroded gully revealed a small source of methane in this type of landscape. Rapid export of carbon-rich soils and an effective drainage system in the gully are likely the main reasons for the limited methane fluxes.

With rising temperatures in Zackenberg Valley, methane emissions are expected to increase drastically during the 21$^{st}$ century. The warming increases permafrost thaw, which could increase surface erosion in the valley. When compared, our findings show the increase in methane emission from undisturbed fen areas has a much larger impact on the valley-wide fluxes than surface erosion. Increased erosion could offset some of the rise in methane fluxes from the valley, but this would require large-scale impacts on vegetated surfaces.

This study shows the importance of multi-year methane monitoring with wide spatial coverage, as interannual
variability is substantial when considering a full composite landscape even in a single valley in the Arctic.

**Data and code availability**

Data from the GEM ClimateBasis and GeoBasis Zackenberg subprograms used in this manuscript are free and
open data, available at https://data.g-e-m.dk/ (registration needed). The data are licensed with terms of use under
the Creative Commons CC-BY-SA license. Processed data and scripts used for the analyses are available from the
715 corresponding author upon reasonable request. Direct links to the GEM data sources are listed here:

AC Water level automatic: https://doi.org/10.17897/mj7b-z461

AC Water level manual: https://doi.org/10.17897/6hcp-m521

Air temperature, 200cm – 60min average: https://doi.org/10.17897/xv96-hc57

Flux monitoring – AC: https://doi.org/10.17897/430p-ds31

Mix-1 Soil moisture: https://doi.org/10.17897/ennb-t831

Snow cover (Central area): https://doi.org/10.17897/499c-h459

Soil temperature, 20cm – 60min average: https://doi.org/10.17897/xw7c-na36

**Author contribution**

JHS, MM, and TRC designed the study, and JHS performed the analyses and created figures and tables. HHC
helped interpreting the geomorphological processes. JHS wrote the manuscript with contributions from all co-
authors.

**Competing interests**

The authors declare that they have no conflict of interest.

**Acknowledgments**

This study was supported by the Faculty of Science and Technology and the thematic centers iCLIMATE and
ARC at Aarhus University. The authors furthermore acknowledge the use of data from the Greenland Ecosystem
Monitoring (GEM) database and are grateful for field support provided by the Zackenberg Research Station.

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
