# Peer review of "Methane in Zackenberg Valley, NE Greenland: Multidecadal growing season fluxes of a high Arctic tundra"

_Biogeosciences, 2021_

## Referee Comment (RC2)

**Reviewer comments: "Methane in Zackenberg Valley, NE Greenland: Multidecadal growing season fluxes of a high Arctic tundra" – Johan J. Scheller et al.**

I thank Scheller et al. for producing an interesting paper. However, although an important topic and a very impressive dataset, the variety of methods used and the results presented do make it slightly hard to follow. I do think the authors could remove some of the repetitiveness with the existing published literature to make it easier to follow. Furthermore, maybe a focus towards the uncertainties found using different methodologies would be a useful addition (rather than just a review of the papers).

The introduction is quite short, therefore there should be some room to expand on the processes linked to the potential increase in methane emissions from Arctic wetlands. At the moment, it really feels like it's missing from the current manuscript.

Although the study on methane emissions from the gully are super interesting and data like this is lacking in the published literature, it gets lost in the sea of all the other data presented. Given the lack of data from this specific study, I don't think it can be a stand-alone paper, but I would put more emphasis on this throughout to make sure it finds its place otherwise it does feel like an add-on.

**Detailed comments**
Line 70 – 71: This line seems repetitive. I would end the introduction on the paragraph beginning on line 67. I would incorporate the use of new data into the paragraph starting line 60.

Section 2.1: I find it hard to follow this section with all the discussion of previous studies and the acronyms used for field sites etc. Could this be paired back and made clearer? Also, your use of Fig. 2a is not clear? Do you mean site (a) in the map?

Could you make the label for the Gully larger? I missed it the first time.

Line 233. You state linear flux model? What type? I think this needs more detail.

Section 3.2: Given the focus of this paper is on the fluxes and your methods section on the details of how this map is produced is very brief, I would move this section either up to the site description or remove to supplementary information alongside Table 2 (which seems unnecessary – either remove from text or remove table).

Section 4.1: I think you could bring in more discussion here about the uncertainties between the different methodologies. This would really strengthen this section.

The presentation of the flux values in the paragraphs in this section make it seem like a results section? You don't really discuss WHY the results may be different? This could be re-written to put more emphasis on why differences were found?

**Figures:**

Figure 3: I wonder if it would be better to make a figure that shows these fluxes in relation to where they are in the landscape? At the moment, from the figure alone I can't tell where Chambers 1 to 6 are located and why we might be seeing differences.

Figure 4: I lost these numbers the first time I looked through the manuscript so they need to be bigger and bolder. Make the dashed line a brighter colour. I think you could present the data for this section more robustly than just a value on a photograph. I would like to see a boxplot to show the variation in fluxes across the 10-day measurement period.

Figure 6: I don't understand this figure unfortunately and I think it could be revised for clarity. Make clearer in caption what the pale green shading represents

Figure 7: What does the inset figure show? Is it just showing the whole year? I'm not sure this is needed.

Figure 7 and 8 have the same figure caption? I presume it is incorrect for Figure 7 given the mention of different marker shapes.

Figure 8: Is this mean $CH_4$ flux represented in this plot?

---

## Author Comment (AC1)

Reviewers' comments to the author: The presented manuscript summarizes chamber-based methane fluxes from the Zackenberg experimental area in Northeastern Greenland. Multiple experiments have been conducted in this area in the period 1997 to present, which are all being considered, while the paper mainly focuses on a data-rich period 2006-2019. Regarding the evaluation of these long time series, the focus has been placed on interannual variability as well as on the extrapolation of fluxes into 2 separate upscaling domains. As a second focus, the paper introduces a new dataset constraining fluxes within a recently formed erosion gully. Based on these new chamber flux data, the authors present a sensitivity study how future erosion events may change net methane emissions within the study area, and how these disturbance effects can be related to expected increases in methane emissions linked to Arctic warming.

The long-term coverage and high temporal frequency of measurements make the Zackenberg experimental area an outstanding resource when it comes to studying carbon cycle processes within the Arctic. This is particularly the case for methane fluxes. Therefore, a study summarizing the wealth of previously reported methane chamber campaigns into a single time series with uniform format is certainly highly valuable. I find the additional focus on the potential effect of gully erosion on landscape scale methane budgets within degrading Arctic landscapes even more interesting. Taken together, the manuscript has a lot to offer, and these topics are certainly of high interest to the readers of this journal. However, I found the weak structuring of this paper to pose quite a hurdle to follow its core message. Also, the authors miss to quantify and discuss several important sources of uncertainty that are essential for supporting their key messages. Main main points of concern are as follows:

*Authors' reply: First, we want to thank you for your throughout comments and constructive criticism. We are confident that your comments will help us improve the manuscript. We see the need to improve both the structure, address sources of uncertainty, and refine several figures for improving their clarity.*

*The remarks from the two reviewers point us toward a revised manuscript with increased emphasis on the sensitivity of the landscape methane flux to future large-scale erosion in Zackenberg Valley while also scaling down the repetitive sections about existing published literature. The comments from the reviewers are in good agreement with each other, and in combination, they chart a clear direction for a carefully revised version of this manuscript.*

*Below are our preliminary replies to the many valuable comments, which we consider carefully in a revised manuscript.*

Reviewers' comments to the author:

1.) if I understood correctly, the upscaled fluxes are based on spatially distributed measurements from the 2007 campaign presented by Tagesson (2013), and interannual variability derived from the automated chamber (AC) program. The latter only covers parts of the land cover types present in the upscaling areas. So you assume that the IAV in these AC systems is representative for changes in the other components that make up the study area. Given the wealth of chamber campaigns that were conducted in the Zackenberg area over the past decades, it must be possible to evaluate this assumption. If not, how do you estimate the uncertainties associated with this approach?

*Authors' reply: We appreciate the comment and possible confusion, and to clarify the approach, we use an alternative, more robust estimate of the landscape fluxes based on all the available data from 2006-2019. This approach combines measurements from:*

- *Highly variable fen fringe (interannual variability derived from the AC)*

- *Measurements from heaths, grasslands, and Salix snowbeds (using data from Tagesson et al., 2013). These values are assumed constant, as they are similar to Christensen et al., 2000 (except for Grasslands).*

- *A linear regression model (unweighted Deming regression) for the fen areas. The simple model enables an estimate of the flux in the fens based on both AC measurements and previous studies. This alternative approach utilizes the existing measurements in the fens and includes the uncertainties from those measurements.*

- *The upscaled fluxes will include the SE from both the flux measurements and the SE from the regression model.*

2.) Related to item 1.), you state in the Discussion that fluxes within constantly wet and dry, resp., areas remain stable over the years, claiming this to be 'indicated by the data from previous site specific campaigns in the valley (l.599)' (btw., such campaigns should thus be perfect to deal with the issue raised above). Next, you state that the bulk of the temporal variability in landscape scale methane fluxes can be attributed to areas with variable wetness levels, and that 'fluxes from different surface classes may respond differently to changes in environmental conditions (l.603f)'. How is this taken into account for the upscaled fluxes presented in this paper? Based on this statement, you either have a static land cover type with highly variable fluxes, or you have land cover types with stable mean flux rates, but variable fractional coverage. In either case, the effect introduces considerable uncertainty into the upscaled product, which must be taken into account and quantified.

*Authors' reply: We can see the need for adjustment, and with the above suggested alternative calculation, we can solve this issue. Using the simplified Hymap surface cover map, we add a 10 m buffer zone along the edges of all fen areas in the valley. These areas are represented by the original six automated chambers, which also cover a gradient of 10 m, the fen fringe. All these boundary areas use this highly variable flux. The remaining fen areas, i.e., those further from the fen fringe, use fluxes from the linear regression model, which relates the variability of the original six chambers and the measured fluxes further out in the fen. The heaths, grasslands, and Salix areas are held constant, as they do not vary much between years (e.g., compared with Christensen et al., 2000). Uncertainties for all surface classes will be present in a revised Figure 5, including their combined uncertainty.*

3.) Your dataset for the erosion gully only covers one single year, and here only a period of 10 days within the late growing season. Even if you break up the anticipated erosion process of the valley floor until 2100 into yearly fragments, how do you cover the long-term development of the eroded surfaces in this concept? I.e., fluxes will follow a specific trajectory as the eroded landscapes slowly approaches a new equilibrium over the decades to follow. This must be taken into account, and properly described in the methods. If you do not have the option to quantify changes in flux rates over the years since disturbance, this feature at least needs to be properly discussed.

*Authors' reply: Thank you for pointing this out. A similar gully in the northern end of the valley developed in 1999, which provides a basis for comparison. The 1999-gully shows regrowth of ~40% over 20 years, equal to 2% per year. This percentage is based on visual interpretation from 100 random points over eroded surfaces in an orthophoto from 2019. This percentage can be added to the projection.*

4) Regarding the prognostic fluxes, it remains undocumented how they were actually derived, with and without erosion:

> what model was used to produce prognostic flux rates?

> how exactly did you estimate the area being affected by erosion in each simulation year, besides considering 25 and 100m erosion corridors?

> You mention that gully formation coincided with the location of ice wedges - was this taken into account when defining areas for future erosion?

> how did you take into consideration that you only had data for that erosion gully within 10 days, and a single observation year?

I find the consideration of the influence of erosion features for the integrated CH4 budget very interesting, but unfortunately one cannot really evaluate the results based on the currently available information.

*Authors' reply: Thank you for letting us know this. We agree that this is a central piece of information to our study, and these points certainly need to be answered. The prognostic flux rates are derived from Geng et al. (2019): they use an exponential fit function to fit temperatures to methane flux, with present and future temperatures forced with the ECHAM climate model. The climate model has a cold bias in the Zackenberg area, so we use the relative increase in methane (equal to +141 %) from modeled present temperatures to modeled 2081-2100 temperatures (RCP8.5).*

*As an alternative to the current sensitivity study, we will include the model SE and base the erosion simulation on three paths. In the first path, we calculate the impact on the mean valley flux if the eroded areas are growing at an annual rate of the same size as the recent gully (720 m2). In the second and third paths, the eroding area starts at 720 m2 per year and grows to 5 and 10 times 720 m2 per year, respectively. The erosion can happen only in areas with excessive ice-rich permafrost near rivers and streams.*

*The observed fluxes from the recent gully agree with the fluxes published in other studies in the area, even though the dataset is limited to 10 days in the late growing season. Even if the full growing season mean flux was ten times larger, the fluxes from the eroded areas would have a minimal effect on the entire valley compared to the uncertainties involved.*

*We believe the alternative calculation will be both more straightforward and document how the prognostic fluxes are derived.*

5.) The summary of datasets from different campaigns over multiple decades is certainty valuable. However, all this material has been published before, and I think that text on this aspect should therefore be reduced within the results part of this manuscript. Besides presenting a summary with a long time series, the main contribution of this paper should rather be to thoroughly discuss the uncertainties that stem from the use of different methodologies over the years, including data processing. The combination of such a heterogeneous dataset may even be subject to systematic biases, so net uncertainties should be a mandatory part of the aggregated time series.

*Authors' reply: We agree that a reduction of already published fluxes is needed, and a discussion of the uncertainties is essential – especially when different datasets are used in the regression model as suggested earlier.*

6.) Regarding the structure, I found several paragraphs and/or display items within the methods section that rather belong into the results, and also a lot of material in the discussion that should actually be part of the methods. Within individual sections, sub-sections jump back and forth between topics. All of this makes it hard to follow the storyline of this manuscript, and should therefore be carefully adjusted. I added several specific recommendations into the detailed comments further below.

*Authors' reply: Thank you for your suggestions on improving the structure and the listed recommendations listed under Minor comments. A revised manuscript will certainly aim at making the adjustments needed for making the structure more streamlined.*

In summary, I think there is a lot of interesting material in this study that makes it worth publishing. At the same time, there are still considerable flaws in the presentation, and many adjustments are required (see major comments above). My recommendation is to reduce the part dealing with the

aggregated chamber flux time series (since it's not based on novel data), and instead put the sensitivity study on gully erosion, and its relative role on upscaled emissions compared to climate change effects, in the foreground. Even though your dataset on the gully fluxes is still limited, an attempt to quantify the impact of such a permafrost degradation would be highly interesting. My overall recommendation is therefore to accept this manuscript for publication, but only after taking care of the major revisions summarized above.

MINOR COMMENTS

INTRODUCTION

- some statements in the first paragraph are currently misleading. At present, the CH4 emissions from the Arctic wetlands do not play a major role for the global CH4 budget. The role of global wetlands is correctly described, but the majority of the emissions can be attributed to tropical regions. The authors should rather focus on the potential emissions from Arctic ecosystems, should permafrost degradation continue, or accelerate, under future climate change

- I think this introduction is missing a paragraph between the current 2nd and 3rd ones that highlights the major scientific uncertainties regarding the Arctic CH4 budget, and underlying processes. I believe your storyline will be more convincing if you first summarize these major problems, and then (in the following paragraph) outline how the presented study addresses (part of) them

- I don't see the need to separate the last 2 sentences as separate paragraphs.

MATERIAL AND METHODS

Section 2.1,

- the section overall is very long. I think this would be better structured if broken up into 2-3 sections

    > description of the actual site (location, land cover, etc.)

    > (recent) climatology: You may consider moving a large fraction of what is currently written about climate/weather to the results section. While I find it appropriate to show mean climate in the methods, here you go into much further detail, showing trends over time, rates of change, etc. If you decide to keep it in here, this may be a part of the site description sub-section, but should follow the landscape description

    > history of observation programs. Very informative, but would be easier to find if listed as a separate section included into sub-section 2.2 (measurements)

- The references to sites shown in Fig.2 are given in a misleading format (e.g. Fig.2a), rather suggesting separate panels. Please use a different format, e.g. (site (a) in Fig.2)

- since one of the study foci is on upscaling, you should add a table in this section that provides the coverage fractions of the main landscape elements within the larger valley floor area, but also within the wetland (moved here from Section 3.2)

- Section 2.2: Merge with later part of Section 2.1, but also with the material in the first few sections of 4.1, to summarize the previous monitoring programs in one place. At the same time, split off the last 3 paragraphs that describe the chamber approach for the gully area into a separate sub-section

- Section 2.2.2: So is what you describe here the map shown as Fig.2 in this work? If so, please reference it properly. If not, please make clear why the remote sensing data needs to be described in detail herein

Section 2.3.4:

- l.233: Please provide some more details on the 'linear flux model'

Section 2.3.5:

- l.247: You claim an increase of CH4 emissions by the end of the century by a factor of 2.43. There is neither a reference nor a method given, so please document where this number came from

RESULTS

Section 3.2

- this information belongs into the methods section. Please move Table 2 into Section 2.1. It's not necessary to repeat these numbers in the text, so the rest of the section can be deleted.

Section 3.3

- the results presentation is a bit weak here. Just plotting the mean fluxes into a photo isn't sufficient to understand the data. It would be helpful to learn more about spatial and temporal variability of this dataset. Did you find consistent flux signals over time at individual plots? Was there a meaningful spatial pattern of flux rates within the gully area?

Section 3.4

- in the way that this is currently presented, I do not see the benefit of showing the temporal variability of upscaled fluxes for these 2 domains. If I got the methodology right, the temporal variation is exactly following those of the AC program, which is shown already in Fig. 3. So why repeat this? Either remove Fig. 3, or find a new format for Figure 5.

Section 3.5

- Figure 6 needs to be revised. It took me a long time, and a lot of scrolling back and forth, to come up with an explanation what might be shown in there. My current interpretation is that the height of all bars indicates the mean valley floor flux WITHOUT erosion. Considering the colors, the red bars show the total mean flux for the valley WITH erosion, and all other colors indicate how this change between both cases can be attributed to erosion within one of the four land cover types. Not sure if this is correct. In any case, please find a new format that emphasizes your intended message. I think it would be easier if you first indicated in the legend that the colors for those 4 LC types indicate changes, not absolute fluxes. Also, it would help if you added a third column within the prognostic scenarios for 'no erosion', and then find a different format to clearly show net fluxes for each erosion scenario.

DISCUSSION

Section 4.1

- Starting l.405, you discuss very broad aspects of spatial and temporal variability in flux rates, and what control factors were identified in previous studies. While this is of course of relevance, obviously these are all previously published results. The main value I see in the current compilation of

summertime flux rates across all these studies is that a long time series is being constructed; however, this comes with additional uncertainties: what is the implication in changes in methodology between studies? Chamber sizes, sampling rates, etc., changed considerably over the years. This should primarily be discussed here.

- Figure 7: I do not see the extra value of the small inset plot in the upper right corner. It is also not documented in the caption. Please remove.

- Figure 7, and the first paragraph of Section 4.1, should be a part of the methods section outlining the previous observation studies summarized in this paper

- l.362-394: This section, including Figure 8, is a result, and nothing is being discussed. So it should be integrated into Section 3. Since basically the same numbers are listed that are given in Figure 8, it's a rather dull read. I recommend transferring the text into a table.

Section 4.2

- l.432f: The explanation that different temporal variability in fCH4 in different sub-section of the fen can be linked to water level fluctuations is plausible. However, it should be straightforward to analyze this quantitatively, since I'm sure that soil moisture and/or water level conditions were closely monitored at each of these automated chamber sites. So why not exploit this dataset?

- starting with line 491, the authors compare their upscaled mean flux rates to other studies across the Arctic. This is certainly interesting, but I think it would make more sense to split this into a separate sub-section of the discussion, and relabel the preceding sections as 'methane flux upscaling', or something along those lines

Section 4.3

- l. 542f: I think this statement should be put into the center of this manuscript!

- l. 546: why do you assume that riverbank erosion will lead to similar effects on fCH4? I would assume that this leads to rather steep cliffs at the river bank, i.e. a very different geomorphology than those shallow gullies depicted in e.g. Figure 4.

- l.558: but there are no uncertainties given for the projected fCH4 values (Figure 6) ..??

- l.562: repeat from statement in l.546, but still no reference …

*Authors' reply: Again, thank you very much for the elaborate comments. We appreciate them a lot. We hope our replies above to your comments show our enthusiasm to improve the contents and structure of the manuscript. The remarks under Minor comments are also a precious input, containing many great observations that will guide us to improving our paper.*

---

## Author Comment (AC2)

Reviewers' comments to the author: I thank Scheller et al. for producing an interesting paper. However, although an important topic and a very impressive dataset, the variety of methods used and the results presented do make it slightly hard to follow. I do think the authors could remove some of the repetitiveness with the existing published literature to make it easier to follow. Furthermore, maybe a focus towards the uncertainties found using different methodologies would be a useful addition (rather than just a review of the papers).

The introduction is quite short, therefore there should be some room to expand on the processes linked to the potential increase in methane emissions from Arctic wetlands. At the moment, it really feels like it's missing from the current manuscript.

Although the study on methane emissions from the gully are super interesting and data like this is lacking in the published literature, it gets lost in the sea of all the other data presented. Given the lack of data from this specific study, I don't think it can be a stand-alone paper, but I would put more emphasis on this throughout to make sure it finds its place otherwise it does feel like an add-on.

*Authors' reply: We would like to thank you for your remarks and specific suggestions for improving the manuscript. Since reading your comments, we see how we can make the paper better. We need to improve both the structure, address sources of uncertainty, and refine several figures for improving their clarity.*

*The remarks from the two reviewers point us toward a revised manuscript with increased emphasis on the sensitivity of the landscape methane flux to future large-scale erosion in Zackenberg Valley while also scaling down the repetitive sections about existing published literature. The comments from the reviewers are in good agreement with each other, and in combination, they chart a clear direction for a carefully revised version of this manuscript.*

*There will be several significant changes to both the upscaling of fluxes from 2006-2019 and major edits to how the sensitivity study is designed in the revised manuscript. We do this to include as much of the measured fluxes as possible while basing even more of the sensitivity study on previously published data. These changes will also help us quantify the sources of error, which enable us to discuss uncertainties more thoroughly. The changes in the calculations will also naturally lead to increasing the emphasis on the methane fluxes from the gully because the gully fluxes play a larger role in the revised sensitivity study. There are more details to these changed calculations in the replies to the other reviewer if you would like to get more information on this at this stage.*

Reviewers' comments to the author:

**Detailed comments**

Line 70 –71: This line seems repetitive. I would end the introduction on the paragraph beginning on line 67. I would incorporate the use of new data into the paragraph starting line 60.

Section 2.1: I find it hard to follow this section with all the discussion of previous studies and the acronyms used for field sites etc. Could this be paired back and made clearer? Also, your use of Fig. 2a is not clear? Do you mean site (a) in the map?

Could you make the label for the Gully larger? I missed it the first time.

Line 233. You state linear flux model? What type? I think this needs more detail.

Section 3.2: Given the focus of this paper is on the fluxes and your methods section on the details of how this map is produced is very brief, I would move this section either up to the site description or remove to supplementary information alongside Table 2 (which seems unnecessary –either remove from text or remove table).

Section 4.1: I think you could bring in more discussion here about the uncertainties between the different methodologies. This would really strengthen this section.

The presentation of the flux values in the paragraphs in this section make it seem like a results section? You don't really discuss WHY the results may be different? This could be re-written to put more emphasis on why differences were found?

*Authors' reply: The detailed comments include a list of suggestions which we agree need some improvements. The questions raised in these comments will be valuable to us when revising this manuscript as soon as possible.*

Reviewers' comments to the author:

**Figures:**

Figure 3: I wonder if it would be better to make a figure that shows these fluxes in relation to where they are in the landscape? At the moment, from the figure alone I can't tell where Chambers 1 to 6 are located and why we might be seeing differences.

Figure 4: I lost these numbers the first time I looked through the manuscript so they need to be bigger and bolder. Make the dashed line a brighter colour. I think you could present the data for this section more robustly than just a value on a photograph. I would like to see a boxplot to show the variation in fluxes across the 10-day measurement period.

Figure 6: I don't understand this figure unfortunately and I think it could be revised for clarity. Make clearer in caption what the pale green shading represents

Figure 7: What does the inset figure show? Is it just showing the whole year? I'm not sure this is needed.

Figure 7 and 8 have the same figure caption? I presume it is incorrect for Figure 7 given the mention of different marker shapes.

Figure 8: Is this mean $CH_4$ flux represented in this plot?

*Authors' reply: We understand that there is a need for improving the figures, including their captions, and we will make sure that these questions will be answered in the revised paper. Again, thank you very much for reviewing our manuscript and the many specific questions, which will help our editing in the coming weeks.*

---

## Author Comment (AC4)

Referee's comments to the author: I thank Scheller et al. for producing an interesting paper. However, although an important topic and a very impressive dataset, the variety of methods used and the results presented do make it slightly hard to follow. I do think the authors could remove some of the repetitiveness with the existing published literature to make it easier to follow. Furthermore, maybe a focus towards the uncertainties found using different methodologies would be a useful addition (rather than just a review of the papers).

The introduction is quite short, therefore there should be some room to expand on the processes linked to the potential increase in methane emissions from Arctic wetlands. At the moment, it really feels like it's missing from the current manuscript.

Although the study on methane emissions from the gully are super interesting and data like this is lacking in the published literature, it gets lost in the sea of all the other data presented. Given the lack of data from this specific study, I don't think it can be a stand-alone paper, but I would put more emphasis on this throughout to make sure it finds its place otherwise it does feel like an add-on.

*Authors' reply: We would like to thank you for your remarks and specific suggestions for improving the manuscript. Since reading your comments, we see how we can make the paper better. We need to improve both the structure, address sources of uncertainty, and refine several figures for improving their clarity.*

*The remarks from the two referees point us toward a revised manuscript with increased emphasis on the sensitivity of the landscape methane flux to future large-scale erosion in Zackenberg Valley while also scaling down the repetitive sections about existing published literature. The comments from the referees are in good agreement with each other, and in combination, they chart a clear direction for a carefully revised version of this manuscript.*

*There will be several significant changes to both the upscaling of fluxes from 2006-2019 and major edits to how the sensitivity study is designed in the revised manuscript. We do this to include as much of the measured fluxes as possible while basing even more of the sensitivity study on previously published data. These changes will also help us quantify the sources of error, enabling a thorough discussion of uncertainties. The changes in the calculations will also naturally lead to increasing the emphasis on the methane fluxes from the gully because the gully fluxes play a more prominent role in the revised sensitivity study.*

Referee's comments to the author:

**Detailed comments**

Line 70 –71: This line seems repetitive. I would end the introduction on the paragraph beginning on line 67. I would incorporate the use of new data into the paragraph starting line 60.

Section 2.1: I find it hard to follow this section with all the discussion of previous studies and the acronyms used for field sites etc. Could this be paired back and made clearer? Also, your use of Fig. 2a is not clear? Do you mean site (a) in the map?

Could you make the label for the Gully larger? I missed it the first time.

Line 233. You state linear flux model? What type? I think this needs more detail.

Section 3.2: Given the focus of this paper is on the fluxes and your methods section on the details of how this map is produced is very brief, I would move this section either up to the site description or remove to supplementary information alongside Table 2 (which seems unnecessary –either remove from text or remove table).

Section 4.1: I think you could bring in more discussion here about the uncertainties between the different methodologies. This would really strengthen this section.

The presentation of the flux values in the paragraphs in this section make it seem like a results section? You don't really discuss WHY the results may be different? This could be re-written to put more emphasis on why differences were found?

*Authors' reply: The detailed comments include a list of suggestions which we agree need some improvements and discussion. The questions raised in these comments will be valuable to us when revising this manuscript as soon as possible.*

Referee's comments to the author:

**Figures:**

Figure 3: I wonder if it would be better to make a figure that shows these fluxes in relation to where they are in the landscape? At the moment, from the figure alone I can't tell where Chambers 1 to 6 are located and why we might be seeing differences.

Figure 4: I lost these numbers the first time I looked through the manuscript so they need to be bigger and bolder. Make the dashed line a brighter colour. I think you could present the data for this section more robustly than just a value on a photograph. I would like to see a boxplot to show the variation in fluxes across the 10-day measurement period.

Figure 6: I don't understand this figure unfortunately and I think it could be revised for clarity. Make clearer in caption what the pale green shading represents

Figure 7: What does the inset figure show? Is it just showing the whole year? I'm not sure this is needed.

Figure 7 and 8 have the same figure caption? I presume it is incorrect for Figure 7 given the mention of different marker shapes.

Figure 8: Is this mean $CH_4$ flux represented in this plot?

*Authors' reply: We understand that there is a need for improving the figures, including their captions, and we will make sure that the questions will be answered in a revised paper. Figure 3 will be omitted from a future revision of the paper, which will allow for a larger emphasis on the landscape fluxes. Figure 4 will have boxplots added, with bolder colors. The calculation behind Figure 6 will be updated, improving its clarity while also allow for uncertainty estimates. An updated Figure 7 will not include the inset figure, and finally, the caption will be double-checked.*

*Again, thank you very much for reviewing our manuscript and the many detailed comments specific questions, which will help our editing in the coming weeks.*

*Final remarks from the authors: We hope our detailed replies to the general comments from the respective referees show a convincing and clear plan for how the manuscript will be revised in relation to the major comments received. In addition, all minor comments and corrections will be accepted and corrections made accordingly. With this we are hoping that the editor agrees to let us proceed with the submission of a revised version of this manuscript.*

---

## Author Response (AR1)

Referee comments 1:

The presented manuscript summarizes chamber-based methane fluxes from the Zackenberg experimental area in Northeastern Greenland. Multiple experiments have been conducted in this area in the period 1997 to present, which are all being considered, while the paper mainly focuses on a data-rich period 2006-2019. Regarding the evaluation of these long time series, the focus has been placed on interannual variability as well as on the extrapolation of fluxes into 2 separate upscaling domains. As a second focus, the paper introduces a new dataset constraining fluxes within a recently formed erosion gully. Based on these new chamber flux data, the authors present a sensitivity study how future erosion events may change net methane emissions within the study area, and how these disturbance effects can be related to expected increases in methane emissions linked to Arctic warming.

The long-term coverage and high temporal frequency of measurements make the Zackenberg experimental area an outstanding resource when it comes to studying carbon cycle processes within the Arctic. This is particularly the case for methane fluxes. Therefore, a study summarizing the wealth of previously reported methane chamber campaigns into a single time series with uniform format is certainly highly valuable. I find the additional focus on the potential effect of gully erosion on landscape scale methane budgets within degrading Arctic landscapes even more interesting. Taken together, the manuscript has a lot to offer, and these topics are certainly of high interest to the readers of this journal. However, I found the weak structuring of this paper to pose quite a hurdle to follow its core message. Also, the authors miss to quantify and discuss several important sources of uncertainty that are essential for supporting their key messages. Main points of concern are as follows:

*Authors' reply: First, we want to thank you for your throughout comments and constructive criticism. Your valuable comments have guided us to improve our manuscript, both with regards to the structure, addressing sources of uncertainty, and refining several figures for improved clarity.*

*The remarks from the two reviewers has pointed us toward a revised manuscript with increased emphasis on uncertainties and differences between datasets and make the study of the landscape methane flux sensitivity to future large-scale erosion in Zackenberg Valley clearer. The comments from the reviewers are in good agreement with each other, and in combination, they charted a clear direction for a carefully revised version of our manuscript.*

*Below are our final replies to the comments, which we considered in our revised manuscript.*

1.) if I understood correctly, the upscaled fluxes are based on spatially distributed measurements from the 2007 campaign presented by Tagesson (2013), and interannual variability derived from the automated chamber (AC) program. The latter only covers parts of the land cover types present in the upscaling areas. So you assume that the IAV in these AC systems is representative for changes in the other components that make up the study area. Given the wealth of chamber campaigns that were conducted in the Zackenberg area over the past decades, it must be possible to evaluate this assumption. If not, how do you estimate the uncertainties associated with this approach?

*Authors' reply: We appreciate the comment and see the lack of clarity. Using an alternative approach, we make a robust estimate of the landscape fluxes, which is on all the available data from 2006-2019. This approach allows us to more improved estimate of errors for the different surface classes. This approach combines measurements from:*

- *Highly variable fen fringe (interannual variability derived from the AC 1-6)*

- *Measurements from heaths, grasslands, and Salix snowbeds (using data from Tagesson et al., 2013, Christensen et al. (2000), and Jørgensen et al. (2015)). These values are assumed constant, as they are changing less in absolute terms between studies.*

- *A linear regression model (unweighted Deming regression) for the fen areas. The simple model enables an estimate of the flux in the fens based on both AC measurements and previous studies. This alternative approach utilizes the existing measurements in the fens and includes the uncertainties from those measurements.*

- *The upscaled fluxes will include the SE from both the flux measurements and the SE from the regression model.*

2.) Related to item 1.), you state in the Discussion that fluxes within constantly wet and dry, resp., areas remain stable over the years, claiming this to be 'indicated by the data from previous site specific campaigns in the valley (l.599)' (btw., such campaigns should thus be perfect to deal with the issue raised above). Next, you state that the bulk of the temporal variability in landscape scale methane fluxes can be attributed to areas with variable wetness levels, and that 'fluxes from different surface classes may respond differently to changes in environmental conditions (l.603f)'. How is this taken into account for the upscaled fluxes presented in this paper? Based on this statement, you either have a static land cover type with highly variable fluxes, or you have land cover types with stable mean flux rates, but variable fractional coverage. In either case, the effect introduces considerable uncertainty into the upscaled product, which must be taken into account and quantified.

*Authors' reply: We can see the need for adjustment, and with the above suggested alternative calculation, we have tried to solve this issue. Using the simplified Hymap surface cover map, we add a 10 m buffer zone along the edges of all fen areas in the valley. These areas are represented by the original six automated chambers, which also cover a gradient of 10 m, the fen fringe. All these boundary areas use this highly variable flux. The remaining fen areas, i.e., those further from the fen fringe, use fluxes from the linear regression model, which relates the variability of the original six chambers to the measured fluxes further out in the fen. The heaths, grasslands, (the new) fell and barren, and Salix areas are held constant, as they do not vary much between years (i.e, between studies from the area). Uncertainties for all surface classes will be present in a revised figure showing the landscape flux, including their combined uncertainty.*

3.) Your dataset for the erosion gully only covers one single year, and here only a period of 10 days within the late growing season. Even if you break up the anticipated erosion process of the valley floor until 2100 into yearly fragments, how do you cover the long-term development of the eroded surfaces in this concept? I.e., fluxes will follow a specific trajectory as the eroded landscapes slowly approaches a new equilibrium over the decades to follow. This must be taken into account, and properly described in the methods. If you do not have the option to quantify changes in flux rates over the years since disturbance, this feature at least needs to be properly discussed.

*Authors' reply: Thank you for pointing this out. A similar gully in the northern end of the valley developed in 1999, which provides a basis for comparison. The 1999-gully shows regrowth of ~40% over 20 years, equal to 2% per year. This percentage is based on visual interpretation from 100 random points over eroded surfaces in an orthophoto from 2019. This percentage can was added to the projection, assuming fluxes similar to those found on undisturbed surfaces near the gully.*

4) Regarding the prognostic fluxes, it remains undocumented how they were actually derived, with and without erosion:

> what model was used to produce prognostic flux rates?

> how exactly did you estimate the area being affected by erosion in each simulation year, besides considering 25 and 100m erosion corridors?

> You mention that gully formation coincided with the location of ice wedges - was this taken into account when defining areas for future erosion?

> how did you take into consideration that you only had data for that erosion gully within 10 days, and a single observation year?

I find the consideration of the influence of erosion features for the integrated CH4 budget very interesting, but unfortunately one cannot really evaluate the results based on the currently available information.

*Authors' reply: Thank you for letting us know this. We agree that this is a central piece of information to our study, and these points certainly need to be answered. The prognostic flux rates are derived from Geng et al. (2019): they use an exponential fit function to fit temperatures to methane flux, with present and future temperatures forced with the ECHAM climate model. The climate model has a cold bias in the Zackenberg area, so we use the relative increase in methane (equal to +141 %) from modeled present temperatures to modeled 2081-2100 temperatures (RCP8.5).*

*In the updated sensitivity study, we include the model SE and base the erosion simulation on three pathways. In the first pathway, we calculate the impact on the mean valley flux if the eroded areas are growing at an annual rate of the same size as the recent gully (720 m2). In the second and third pathways, the eroding area starts at 720 m2 per year and grows to 5 and 10 times 720 m2 per year, respectively. The erosion can happen only in areas with excessive ice-rich permafrost near rivers and streams.*

*The observed fluxes from the recent gully agree with the fluxes published in other studies in the area, even though the dataset is limited to 10 days in the late growing season. The methane flux was limited in the gully itself, but single measurements showed slightly higher emissions in recently eroded areas. The gully contained a silty soil mixture with assumed limited potential for methane production.*

5.) The summary of datasets from different campaigns over multiple decades is certainty valuable. However, all this material has been published before, and I think that text on this aspect should therefore be reduced within the results part of this manuscript. Besides presenting a summary with a long time series, the main contribution of this paper should rather be to thoroughly discuss the uncertainties that stem from the use of different methodologies over the years, including data processing. The combination of such a heterogeneous dataset may even be subject to systematic biases, so net uncertainties should be a mandatory part of the aggregated time series.

*Authors' reply: We agree that a reduction of already published fluxes is needed, and a discussion of the uncertainties is essential – especially when different datasets are used in the regression model as suggested earlier. In the revised version of the manuscript, results from previous studies are limited, and we discuss instead the reasons for differences between the results and the associated uncertainties.*

6.) Regarding the structure, I found several paragraphs and/or display items within the methods section that rather belong into the results, and also a lot of material in the discussion that should actually be part of the methods. Within individual sections, sub-sections jump back and forth between topics. All of this makes it hard to follow the storyline of this manuscript, and should therefore be carefully adjusted. I added several specific recommendations into the detailed comments further below.

*Authors' reply: Thank you for your suggestions on improving the structure and the listed recommendations listed under Minor comments. In our revised manuscript, we have done our best to making the structure more streamlined.*

In summary, I think there is a lot of interesting material in this study that makes it worth publishing. At the same time, there are still considerable flaws in the presentation, and many adjustments are required (see major comments above). My recommendation is to reduce the part dealing with the aggregated chamber flux time series (since it's not based on novel data), and instead put the

sensitivity study on gully erosion, and its relative role on upscaled emissions compared to climate change effects, in the foreground. Even though your dataset on the gully fluxes is still limited, an attempt to quantify the impact of such a permafrost degradation would be highly interesting. My overall recommendation is therefore to accept this manuscript for publication, but only after taking care of the major revisions summarized above.

MINOR COMMENTS

INTRODUCTION

- some statements in the first paragraph are currently misleading. At present, the CH4 emissions from the Arctic wetlands do not play a major role for the global CH4 budget. The role of global wetlands is correctly described, but the majority of the emissions can be attributed to tropical regions. The authors should rather focus on the potential emissions from Arctic ecosystems, should permafrost degradation continue, or accelerate, under future climate change

*Authors' reply: We have edited the wording so that the emphasis is now on potential future emissions.*

- I think this introduction is missing a paragraph between the current 2nd and 3rd ones that highlights the major scientific uncertainties regarding the Arctic CH4 budget, and underlying processes. I believe your storyline will be more convincing if you first summarize these major problems, and then (in the following paragraph) outline how the presented study addresses (part of) them.

*Authors' reply: We have added a paragraph (l. 58-69) that highlights these uncertainties.*

- I don't see the need to separate the last 2 sentences as separate paragraphs.

*Authors' reply: The sentences are combined into one paragraph.*

MATERIAL AND METHODS

Section 2.1,

- the section overall is very long. I think this would be better structured if broken up into 2-3 sections

> description of the actual site (location, land cover, etc.)

> (recent) climatology: You may consider moving a large fraction of what is currently written about climate/weather to the results section. While I find it appropriate to show mean climate in the methods, here you go into much further detail, showing trends over time, rates of change, etc. If you decide to keep it in here, this may be a part of the site description sub-section, but should follow the landscape description

> history of observation programs. Very informative, but would be easier to find if listed as a separate section included into sub-section 2.2 (measurements)

*Authors' reply: The section is split into two: 2.1.1 Study area, and 2.1.2 Climatology. The detailed description of environmental data is moved to section 2.3.1. The history of measurements is now in its own subsection 2.2.1 under 2.2 Methane flux measurements.*

- The references to sites shown in Fig.2 are given in a misleading format (e.g. Fig.2a), rather suggesting separate panels. Please use a different format, e.g. (site (a) in Fig.2)

*Authors' reply: The references to the map is now changed to, e.g., (Fig. 1, site a).*

- since one of the study foci is on upscaling, you should add a table in this section that provides the coverage fractions of the main landscape elements within the larger valley floor area, but also within the wetland (moved here from Section 3.2)

*Authors' reply: The table containing coverage fractions is moved to 2.1.1 Study area*

- Section 2.2: Merge with later part of Section 2.1, but also with the material in the first few sections of 4.1, to summarize the previous monitoring programs in one place. At the same time, split off the last 3 paragraphs that describe the chamber approach for the gully area into a separate sub-section

*Authors' reply: The gully measurements are now described in their own subsection, and descriptions of the monitoring programs are summarized in one place, including the figure (new Fig. 2) showing an overview of measurement periods.*

- Section 2.2.2: So is what you describe here the map shown as Fig.2 in this work? If so, please reference it properly. If not, please make clear why the remote sensing data needs to be described in detail herein

*Authors' reply: The section has been deleted and referenced under the new Fig. 1.*

Section 2.3.4:

- l.233: Please provide some more details on the 'linear flux model'

*Authors' reply: We added the description: "...ordinary least square linear (OLS) regression described..."*

Section 2.3.5:

- l.247: You claim an increase of CH4 emissions by the end of the century by a factor of 2.43. There is neither a reference nor a method given, so please document where this number came from

*Authors' reply: We added details to the description, the wording can now be found on l. 265-275.*

RESULTS

Section 3.2

- this information belongs into the methods section. Please move Table 2 into Section 2.1. It's not necessary to repeat these numbers in the text, so the rest of the section can be deleted.

*Authors' reply: The section is omitted from the revised manuscript.*

Section 3.3

- the results presentation is a bit weak here. Just plotting the mean fluxes into a photo isn't sufficient to understand the data. It would be helpful to learn more about spatial and temporal variability of this dataset. Did you find consistent flux signals over time at individual plots? Was there a meaningful spatial pattern of flux rates within the gully area?

*Authors' reply: Three boxplots have been added to the figure, along with a description of both temporal and spatial patterns in data.*

Section 3.4

- in the way that this is currently presented, I do not see the benefit of showing the temporal variability of upscaled fluxes for these 2 domains. If I got the methodology right, the temporal variation is exactly following those of the AC program, which is shown already in Fig. 3. So why repeat this? Either remove Fig. 3, or find a new format for Figure 5.

*Authors' reply: The figure showing variability at AC has been removed, and the upscaled figure has been changed. Although the plots from Rylekærene and the valley floor look similar, there is an important point in keeping both domains – the larger study area contains a larger fraction of areas that reduces the landscape flux (heaths, and Salix snowbeds, and fell and barren), but this is clearer in the new Fig. 6 in the revised manuscript.*

Section 3.5

- Figure 6 needs to be revised. It took me a long time, and a lot of scrolling back and forth, to come up with an explanation what might be shown in there. My current interpretation is that the height of all bars indicates the mean valley floor flux WITHOUT erosion. Considering the colors, the red bars show the total mean flux for the valley WITH erosion, and all other colors indicate how this change between both cases can be attributed to erosion within one of the four land cover types. Not sure if this is correct. In any case, please find a new format that emphasizes your intended message. I think it would be easier if you first indicated in the legend that the colors for those 4 LC types indicate changes, not absolute fluxes. Also, it would help if you added a third column within the prognostic scenarios for 'no erosion', and then find a different format to clearly show net fluxes for each erosion scenario.

*Authors' reply: In the new Fig. 7, we added 'no erosion' along with the calculated valley fluxes from Fig. 6 for comparison. The average fluxes are now shown as annotation on the figure itself, which we hope will make the interpretation clearer for the reader. The impacts from different land cover classes was omitted, as the entire calculation was changed.*

DISCUSSION

Section 4.1

- Starting l.405, you discuss very broad aspects of spatial and temporal variability in flux rates, and what control factors were identified in previous studies. While this is of course of relevance, obviously these are all previously published results. The main value I see in the current compilation of summertime flux rates across all these studies is that a long time series is being constructed; however, this comes with additional uncertainties: what is the implication in changes in methodology between studies? Chamber sizes, sampling rates, etc., changed considerably over the years. This should primarily be discussed here.

*Authors' reply: We changed the focus of the discussion here to include comparisons of the methods and possible reasons for differences in uncertainties and fluxes.*

- Figure 7: I do not see the extra value of the small inset plot in the upper right corner. It is also not documented in the caption. Please remove.

*Authors' reply: The insert has been removed in the new Fig 2.*

- Figure 7, and the first paragraph of Section 4.1, should be a part of the methods section outlining the previous observation studies summarized in this paper

*Authors' reply: The figure has been moved to section 2.2.1 along with the first paragraph.*

- l.362-394: This section, including Figure 8, is a result, and nothing is being discussed. So it should be integrated into Section 3. Since basically the same numbers are listed that are given in Figure 8, it's a rather dull read. I recommend transferring the text into a table.

*Authors' reply: The figure has been moved to section 3.2 with a brief text. The text was not transferred into a table, because the figure contains all the information already.*

Section 4.2

- l.432f: The explanation that different temporal variability in fCH4 in different sub-section of the fen can be linked to water level fluctuations is plausible. However, it should be straightforward to analyze this quantitatively, since I'm sure that soil moisture and/or water level conditions were closely monitored at each of these automated chamber sites. So why not exploit this dataset?

*Authors' reply: Unfortunately, this is not as straightforward as one could have hoped. There is only data available from two of the ten chambers, and water level measurements are only available as daily measurements in first years of the time series in one chamber. No water level measurements are available from the outermost group of chambers (chambers 7 to 10). For this reason, a quantitative analysis is limited, but we discuss the available data (l. 540 to 557).*

- starting with line 491, the authors compare their upscaled mean flux rates to other studies across the Arctic. This is certainly interesting, but I think it would make more sense to split this into a separate sub-section of the discussion, and relabel the preceding sections as 'methane flux upscaling', or something along those lines

*Authors' reply: The section has been split and can be found under 4.4 Methane flux upscaling in Arctic landscapes.*

Section 4.3

- l. 542f: I think this statement should be put into the center of this manuscript!

*Authors' reply: In the revised manuscript, the proximity of rivers and ice wedge areas are central components for the calculation of erosion pathways, which gives the statement a more central role in the manuscript.*

- l. 546: why do you assume that riverbank erosion will lead to similar effects on fCH4? I would assume that this leads to rather steep cliffs at the river bank, i.e. a very different geomorphology than those shallow gullies depicted in e.g. Figure 4.

*Authors' reply: The revised method does not rely on the riverbank erosion, but only on erosion similar to the gully.*

- l.558: but there are no uncertainties given for the projected fCH4 values (Figure 6) ..??

*Authors' reply: The revised version now includes uncertainties for the projections.*

- l.562: repeat from statement in l.546, but still no reference …

*Authors' reply: A reference to Fig. 5 was added along with a specification of 'These findings'*

*Authors' reply: Again, thank you very much for the elaborate comments. We appreciate them a lot. We hope our replies above to your comments illustrate how we have revised the manuscript. The remarks under Minor comments are also a precious input, containing many great observations that guided us to improve our paper.*

Referee comments 2:

I thank Scheller et al. for producing an interesting paper. However, although an important topic and a very impressive dataset, the variety of methods used and the results presented do make it slightly hard to follow. I do think the authors could remove some of the repetitiveness with the existing published literature to make it easier to follow. Furthermore, maybe a focus towards the uncertainties found using different methodologies would be a useful addition (rather than just a review of the papers).

The introduction is quite short, therefore there should be some room to expand on the processes linked to the potential increase in methane emissions from Arctic wetlands. At the moment, it really feels like it's missing from the current manuscript.

Although the study on methane emissions from the gully are super interesting and data like this is lacking in the published literature, it gets lost in the sea of all the other data presented. Given the lack of data from this specific study, I don't think it can be a stand-alone paper, but I would put more emphasis on this throughout to make sure it finds its place otherwise it does feel like an add-on.

*Authors' reply: We would like to thank you for your remarks and specific suggestions for improving the manuscript. Since reading your comments, we have done our best to make the paper better. We have improved both the structure, addressed sources of uncertainty, and refined several figures for better clarity.*

**Detailed comments**

Line 70 –71: This line seems repetitive. I would end the introduction on the paragraph beginning on line 67. I would incorporate the use of new data into the paragraph starting line 60.

*Authors' reply: We have adjusted the revised manuscript so that it incorporates the three suggestions above.*

Section 2.1: I find it hard to follow this section with all the discussion of previous studies and the acronyms used for field sites etc. Could this be paired back and made clearer? Also, your use of Fig. 2a is not clear? Do you mean site (a) in the map?

*Authors' reply: We have restructured the section to make it easier to follow. References to the sites in the figure now has the following structure: (Fig. 1, site a)*

Could you make the label for the Gully larger? I missed it the first time.

*Authors' reply: The labels are a little larger now and the contrast has been increased, which makes the text stand out from the background more.*

Line 233. You state linear flux model? What type? I think this needs more detail.

*Authors' reply: We added the brief description: "…ordinary least square linear (OLS) regression described…"*

Section 3.2: Given the focus of this paper is on the fluxes and your methods section on the details of how this map is produced is very brief, I would move this section either up to the site description or remove to supplementary information alongside Table 2 (which seems unnecessary –either remove from text or remove table).

*Authors' reply: We added the description: The section has been removed from the revised version.*

Section 4.1: I think you could bring in more discussion here about the uncertainties between the different methodologies. This would really strengthen this section.

*Authors' reply: In the discussion, we have increased the emphasis on uncertainties, including differences between sites and timing.*

The presentation of the flux values in the paragraphs in this section make it seem like a results section? You don't really discuss WHY the results may be different? This could be re-written to put more emphasis on why differences were found?

*Authors' reply: We have added emphasis on possible reasons for the differences, and flux values have been moved to the results section.*

**Figures:**

Figure 3: I wonder if it would be better to make a figure that shows these fluxes in relation to where they are in the landscape? At the moment, from the figure alone I can't tell where Chambers 1 to 6 are located and why we might be seeing differences.

*Authors' reply: The figure has been deleted because it shows the same variability as the landscape flux figure (new Fig. 6).*

Figure 4: I lost these numbers the first time I looked through the manuscript so they need to be bigger and bolder. Make the dashed line a brighter colour. I think you could present the data for this section more robustly than just a value on a photograph. I would like to see a boxplot to show the variation in fluxes across the 10-day measurement period.

*Authors' reply: The figure has been updated with bolder colors matching the surface class colors on e.g., the study area map (new Fig. 1), and with a boxplot for each class.*

Figure 6: I don't understand this figure unfortunately and I think it could be revised for clarity. Make clearer in caption what the pale green shading represents

*Authors' reply: The figure (new Fig. 7) has been changed significantly, which should make it clearer.*

Figure 7: What does the inset figure show? Is it just showing the whole year? I'm not sure this is needed.

*Authors' reply: The inset figure has been deleted (new Fig. 2)*

Figure 7 and 8 have the same figure caption? I presume it is incorrect for Figure 7 given the mention of different marker shapes.

*Authors' reply: Yes, it is incorrect. It has now been fixed.*

Figure 8: Is this mean $CH_4$ flux represented in this plot?

*Authors' reply: Mean fluxes for each dataset is shown per year, but the mean landscape fluxes are not shown (new Fig. 4)*

*Again, thank you very much for reviewing our manuscript and the many detailed comments specific questions, which will helped us a lot for the revision of our manuscript.*

*Final remarks from the authors: We hope our detailed replies to the general comments from the respective referees show how the manuscript has be revised in relation to the major comments received. In addition, all minor comments and corrections has been accepted and corrections made accordingly.*